# Proton signaling links epithelial sensing to neural control of host defense in *C. elegans*

Ying Lei [1,4], Xu Zhan[1,4], Chao Chen [2], Yuxin Liu[1,3], Ying Wang[1] & Ping Liu [1] ✉

Effective host defense against pathogens requires coordinated behavioral and immune responses, yet the mechanisms that couple epithelial sensing to these systemic defenses remain poorly understood. Here, we identify a proton-mediated gut-to-neuron signaling pathway that orchestrates host defense in *C. elegans*. Intestinal pathogens stimulate mechanosensitive $Ca^{2+}$ influx into intestinal epithelial cells (IECs) through the TRP channel GON-2, activating the $Na^+/H^+$ exchanger NHX-6 via the calmodulin CMD-1 to drive basolateral proton release. These protons activate cholinergic motor neurons through the acid-sensing ion channel ASIC-1, enhancing cholinergic transmission to promote both pathogen avoidance and intestinal innate immunity. Notably, mouse NHE1 and ASIC1a can functionally substitute for their nematode counterparts. Together, these findings demonstrate a role for proton signaling in gut-to-neuron communication, revealing a potentially conserved mechanism that links epithelial sensing to neuroimmune defense.

Animals rely on both behavioral and immune mechanisms to defend against pathogen infections[1–3]. While immune responses eliminate invading pathogens, behavioral responses help prevent encounters, promote escape, and reduce further ingestion[1–4]. The coordination of these two mechanisms is essential for effective pathogen defense, with the nervous system serving as a key integrative regulator[5–7]. Specifically, the initiation of avoidance behavior relies on neural detection of sensory cues from external and ingested pathogens[1,2,4], while neuroimmune communication fine-tunes immune responses and maintains systemic homeostasis[5–7]. This bidirectional crosstalk can also influence neuronal functions, leading to behavioral changes[7–9]. However, the molecular mechanisms that coordinate behavioral and immune defenses via this bidirectional communication remain poorly defined.

Intestinal epithelial cells (IECs) form a critical anatomical and functional barrier against ingested pathogens while closely interacting with the nervous system[9–11]. In mammals, IECs consist of specialized subtypes that communicate with the luminal environment[11]. For example, Paneth cells sense intestinal pathogens and release antimicrobial substances, contributing to innate immune defenses[12,13].

Enteroendocrine cells (EECs), acting as chemosensors, detect luminal contents and secrete neuroendocrine molecules[10,14,15]. In the mucosa, the vagus nerve is positioned near EECs and expresses receptors for gut-derived molecules, allowing basolaterally released signals to reach the brain and influence behavior[4,10,14,15]. Conversely, the vagus nerve releases acetylcholine (ACh)[7,11]. Activation of muscarinic ACh receptors (mAChRs) induces secretion from Paneth cells and EECs[11]. This apical-basolateral dichotomy and bidirectional neuroepithelial communication position IECs to coordinate intestinal immune responses with organism-wide behavioral defenses. However, the specific molecular pathways by which IECs communicate with neurons to coordinate host defense remain poorly understood.

The nematode *C. elegans* provides a powerful model for dissecting such inter-tissue signaling and neuroimmune communication at molecular, single-cell, and whole-animal levels[2,5,16,17]. In the laboratory, *C. elegans* feeds on non-pathogenic *E. coli* OP50, and intestinal infections can be readily introduced with human pathogens, facilitating mechanistic studies[17]. Despite its simple anatomy, *C. elegans* is capable of behaviorally avoiding and immunologically eliminating ingested pathogens[18–25]. Its innate immune pathways, including PMK-1/p38

[1]Department of Pathophysiology, Key Laboratory of Ministry of Education of China and Hubei Province for Neurological Disorders, School of Basic Medicine, Tongji Medical College, Huazhong University of Science and Technology, Wuhan, Hubei, China. [2]Department of Orthopaedics, Union Hospital, Tongji Medical College, Huazhong University of Science and Technology, Wuhan, Hubei, China. [3]Department of Pathology, Xinzhou District People's Hospital, Wuhan, Hubei, China. [4]These authors contributed equally: Ying Lei, Xu Zhan. ✉e-mail: pingl@hust.edu.cn

MAPK and canonical Wnt signaling, are evolutionarily conserved[21,26,27]. Like mammals, its nervous system regulates intestinal immunity, and intestine-to-neuron signaling can modulate pathogen avoidance[16,19,21,28–31]. Several neuronal regulators of behavioral and immune responses have been identified[20,30,32–36], yet whether a conserved intestine-to-neuron signaling pathway exists to coordinate host defense remains unknown.

Here, we identify a proton-mediated gut-to-neuron signaling pathway that links epithelial sensing to both behavioral and immune defenses in *C. elegans*. Intestinal pathogens stimulate mechanosensitive $Ca^{2+}$ influx into IECs via the TRP channel GON-2, activating the $Na^+/H^+$ exchanger (NHE) NHX-6 through CMD-1/calmodulin to drive basolateral proton release. These protons activate cholinergic ventral nerve cord (VNC) motor neurons via the acid-sensing ion channel (ASIC) ASIC-1, enhancing cholinergic transmission to promote pathogen avoidance. Concurrently, cholinergic signaling acts back on IECs to activate canonical Wnt and PMK-1/p38 MAPK pathways via the mAChR GAR2;GAR-3, enhancing intestinal immunity. Using the *C. elegans* intestine-motor neuron unit as a platform, we show that mouse NHE1 and ASIC1a can functionally substitute for their nematode counterparts.

Together, these findings demonstrate a role for protons as gut-derived signaling molecules that couple epithelial stress to neuronal and immune responses, revealing a potentially conserved mechanism that coordinates host defense against intestinal pathogens.

## Results

### ASIC-1 mediates the acid sensitivity of VA and VB motor neurons in *C. elegans*

Acid-sensing ion channels (ASICs), members of the degenerin/epithelial $Na^+$ channel (DEG/ENaC) family, are evolutionarily conserved from *C. elegans* to mammals, including humans and mice[37,38]. We previously found that *C. elegans* ASIC-1 functions as a mechanoreceptor in VA and VB motor neurons[39]. However, mammalian ASICs are primarily recognized as proton-gated cation channels that respond to extracellular acidic pH[38]. Consistent with this, *C. elegans* ASIC-1 can form proton-gated homomeric channels in *Xenopus* oocytes[40].

To assess whether VA and VB neurons are acid-sensitive, we applied acidic pH solutions directly to the VNC of dissected worms and monitored neuronal responses by calcium imaging. We used a transgenic strain expressing GCaMP6 under the *unc-17Δ1* promoter[39] (*Punc-17Δ1*), which drives expression in multiple types of cholinergic VNC motor neurons, including A-type (VA and DA), B-type (VB and DB), and AS-type neurons. Upon acidic pH stimulation, all VA and VB in the imaging field showed robust calcium responses (Fig. 1a, b). Unexpectedly, DA and DB also exhibited acid-evoked responses, although weaker than those observed in VA and VB (Fig. 1b). In contrast, AS neurons showed no detectable response (Fig. 1b). Acid-evoked responses in A- and B-type motor neurons were pH-dependent, becoming detectable at pH 6 and plateauing at pH 4 (Fig. 1a, b). Thus, we selected pH 4 for further analysis.

Since A- and B-type motor neurons are synaptically interconnected and receive inputs from other neurons[41], we next investigated whether their responses were intrinsic. Disruption of gap junction proteins expressed in VA/VB (INX-3, INX-12, and UNC-9)[42] or in DA/DB (INX-3, INX-7, and UNC-9)[42] had no effect on acid-evoked responses (Supplementary Fig. 1a). Responses in VA and VB also persisted in both *unc-13(e51)* and *unc-31(e169)* mutants (Fig. 1c and Supplementary Fig. 1b), which are defective in neurotransmitter and neuropeptide release, respectively[43,44]. In contrast, responses in DA and DB were abolished in *unc-13(e51)* mutants (Fig. 1c and Supplementary Fig. 1b). Furthermore, inhibiting synaptic release from VA and VB using tetanus toxin (TeTx) under *Pdel-1*[39] significantly reduced acid-evoked responses in DA and DB (Fig. 1c and Supplementary Fig. 1b). Although additional innexins may be expressed in these neurons[45,46],

and RNAi efficiency and cell-type specificity were not directly assessed, the data do not support a major contribution of gap junctions. Together, these results suggest that VA and VB are intrinsically acid-sensitive, whereas the responses observed in DA and DB are likely due to synaptic input from VA and VB.

We then examined whether ASIC-1 mediates the acid sensitivity of VA and VB. In *asic-1(ok415)* deletion mutants, acid-evoked responses in A- and B-type motor neurons were nearly absent (Fig. 1d and Supplementary Fig. 1c). Expression of wild-type *asic-1* in VA and VB under *Pdel-1* fully rescued acid-evoked responses in VA and VB, as well as in DA and DB (Fig. 1d and Supplementary Fig. 1c). In contrast, *asic-1(ok415)* had no effect on the voltage-dependent whole-cell currents of VA and VB (Supplementary Fig. 1d, e), consistent with our previous findings[39]. Together, these results demonstrate that ASIC-1 mediates the acid sensitivity of VA and VB without generally altering their ion channel function, and further support the notion that acid-evoked responses in DA and DB arise from VA and VB synaptic input.

### Intestinal pathogens induce proton secretion from IECs through NHX-6

Next, we investigated the source of protons that may act on VA and VB in vivo. The *C. elegans* intestine runs alongside the VNC across the body cavity (Fig. 2a) and is surrounded by the commissures of A- and B-type motor neurons[47]. In mammals, intestinal pathogens or inflammation can cause acidosis in the basolateral space of IECs[48,49]. To test this possibility in *C. elegans*, we visualized the extracellular pH of A- and B-type motor neurons during pathogen infections using pHluorin, a pH-sensitive GFP reporter whose fluorescence decreases as pH reduces[50]. We fused pHluorin to the extracellular side of the mouse CD8 transmembrane domain[51] and expressed the fusion protein in cholinergic VNC motor neurons using *Punc-17Δ1* (Fig. 2a). Two paradigmatic human pathogenic bacteria, the Gram-positive *S. aureus* and the Gram-negative *P. aeruginosa*, were used to infect the transgenic worms. Compared to control worms fed on *E. coli*, those infected with *S. aureus* or *P. aeruginosa* displayed significantly reduced pHluorin fluorescence, indicating decreased extracellular pH (Fig. 2b). Pathogen infections induced extracellular acidification at multiple motor neuron locations along VNC (Fig. 2b and Supplementary Fig. 3a). In contrast, neither heat-killed *S. aureus* nor the nonpathogenic mutant *P. aeruginosa-gacA(-)* affected pHluorin fluorescence (Fig. 2b), arguing against contributions from bacterial identity or dietary differences. Although these experiments do not directly determine whether pathogen infections cause global pseudocoelomic acidification, they demonstrate that the extracellular space surrounding motor neurons at widespread VNC sites becomes acidified during pathogen infections. Across these conditions, the co-expressed, pH-insensitive mStrawberry signal remained unchanged during pathogen exposure (Supplementary Fig. 4a), indicating stable reporter expression.

We next examined the role of NHEs in proton secretion, as these transporters are known to acidify the extracellular environment by releasing protons[50,52]. Among the nine *C. elegans* NHEs, NHX-1, NHX-2, NHX-6, and NHX-7 are expressed predominantly in the intestine, with NHX-6 and NHX-7 localized to the basolateral membrane of IECs[52]. Previous studies have shown that NHX-7 is necessary for phasic proton release from posterior IECs to stimulate posterior body muscle contraction during defecation[50,53]. Nevertheless, we tested all NHEs using either available mutants or IEC-specific RNAi worms. Only *nhx-6(ok609)* deletion mutants exhibited significantly increased pHluorin fluorescence during pathogen infections (Fig. 2c and Supplementary Figs. 2 and 3b). This result aligns with the broader expression of NHX-6 in the intestine compared to the posterior restriction of NHX-7[50,52]. The increased pHluorin fluorescence in *nhx-6(ok609)* mutants was rescued by expressing wild-type *nhx-6* specifically in IECs using *Pges-1*[54] (Fig. 2c and Supplementary Fig. 3b). mStrawberry fluorescence was unaffected by pathogen exposure across these genotypes (Supplementary

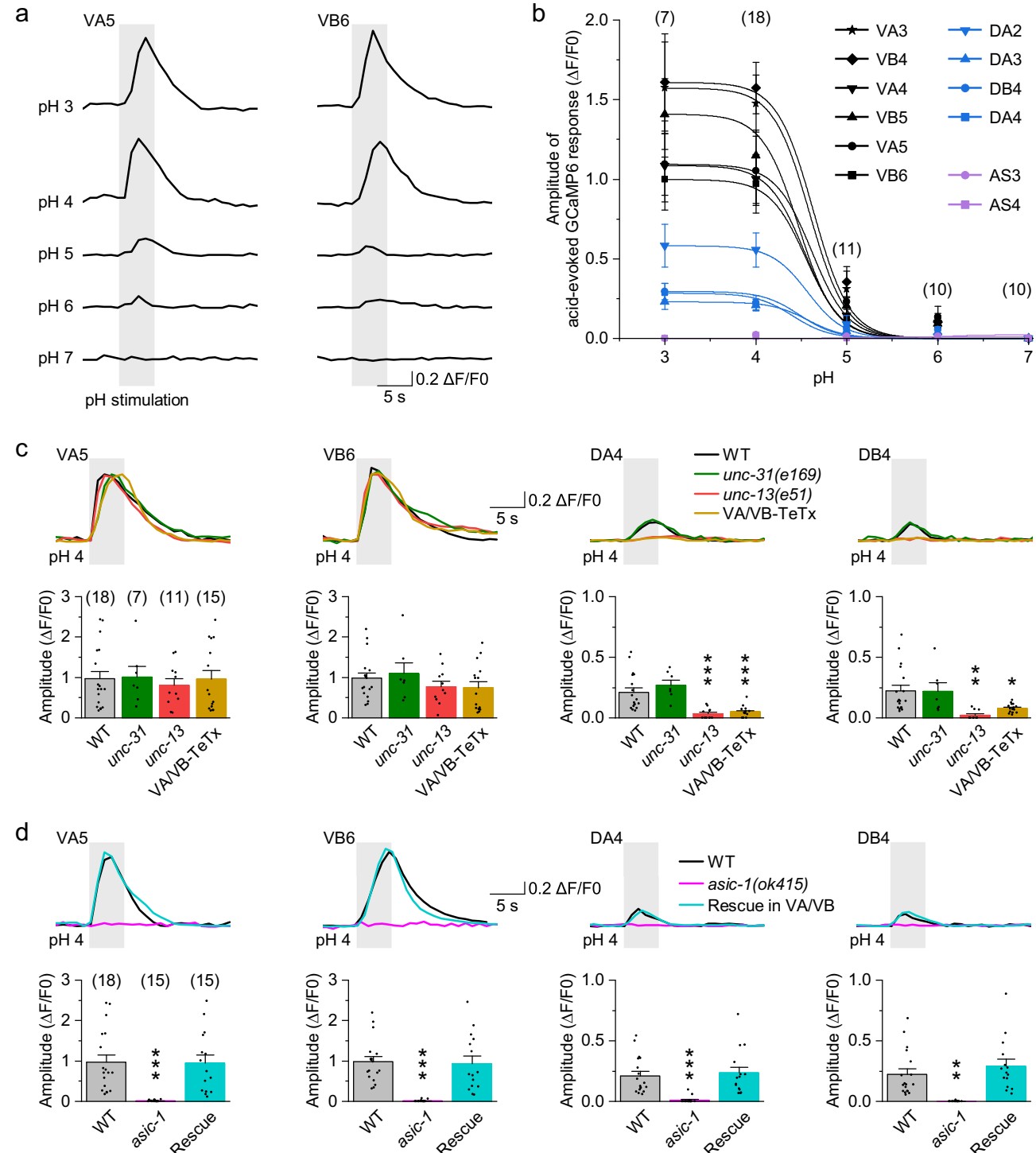

**Fig. 1 | ASIC-1 mediates the acid sensitivity of VA and VB motor neurons.**
**a** Representative calcium responses of VA5 and VB6 to acidic pH stimulation. Gray bars indicate 5-s acidic pH stimulation. A transgenic strain expressing GCaMP6 in cholinergic VNC motor neurons under *Punc-17Δ1* was used. **b** Relationship between acid-evoked GCaMP6 amplitude in cholinergic VNC motor neurons and pH. Data were fitted using the Boltzmann function, $y = A2 + (A1 − A2)/(1 + \exp((x − x0)/dx))$. **c**, **d** pH 4-evoked calcium responses in A- and B-type motor neurons of the indicated genotypes. Top, sample traces. Bottom, comparisons of pH 4-evoked GCaMP6

amplitude. WT, wild type. VA/VB-TeTx, transgenic worms expressing tetanus toxin (TeTx) in VA and VB using *Pdel-1*. Rescue was done by expressing wild-type *asic-1* in VA and VB using *Pdel-1*. $p = 0.9995$, 0.93515, 0.99992, 0.96515, 0.78204, 0.64539, 0.63149, 0.0004, 0.0003, 0.99997, 0.00148, and 0.0176 (**c**), and 0.0003, 0.99452, <0.0001, 0.97522, 0.0005, 0.89762, 0.00131, and 0.48041 (**d**). *$p < 0.05$, **$p < 0.01$, and ***$p < 0.001$ (one-way ANOVA with Tukey's post hoc test). Brackets contain the number of animals tested (*n*). Data are shown as means ± SEM. Source data are provided as a Source data file.

Fig. 4b). Together, these results suggest that intestinal pathogens stimulate tonic proton release from IECs via NHX-6, leading to acidification of the extracellular environment surrounding VNC motor neurons.

**Pathogen-induced IEC proton secretion requires TRP channel GON-2-mediated Ca²⁺ influx and CMD-1/calmodulin**
In mammals, NHE activity is regulated by Ca²⁺/calmodulin signaling[50,55]. Many innate immune cells express the mechanosensitive ion channels

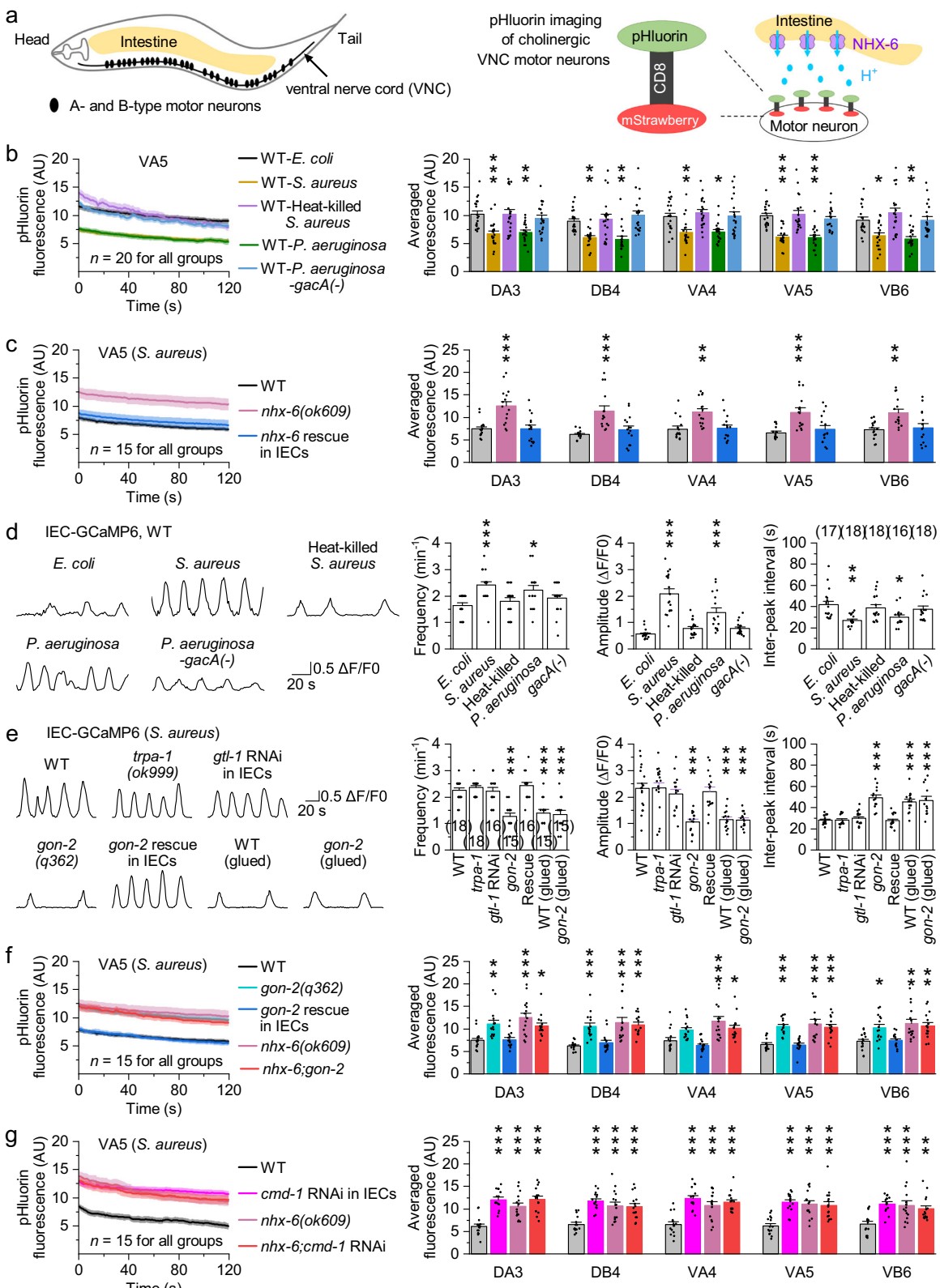

TRPV4, which mediates Ca²⁺ influx upon activation[56]. In *C. elegans*, the TRP channel GTL-2 mediates Ca²⁺ influx in response to membrane damage in epidermal cells[57]. To determine whether pathogens alter Ca²⁺ dynamics in IECs, we monitored cytosolic Ca²⁺ using a transgenic strain expressing GCaMP6 under *Pges-1*. In wild type fed *E. coli*, we observed frequent spontaneous Ca²⁺ transients in IECs (Fig. 2d). Infection with either *S. aureus* or *P. aeruginosa* significantly increased

the frequency and amplitude of Ca²⁺ transients and shortened inter-peak intervals, whereas heat-killed *S. aureus* or *P. aeruginosa-gacA(-)* had no effect (Fig. 2d). Among the three TRP channels expressed in IECs, TRPA-1, GTL-1, and GON-2[58,59], only *gon-2(q362)* mutants exhibited a strong reduction in pathogen-induced Ca²⁺ transients, which was rescued by IEC-specific expression of wild-type *gon-2* under *Pges-1* (Fig. 2e and Supplementary Fig. 3c). In contrast, *trpa-1(ok999)* mutants

**Fig. 2 | Pathogen-induced IEC proton secretion requires TRP channel GON-2-mediated Ca²⁺ influx and CMD-1/calmodulin. a** Diagrams depicting the anatomical locations of the intestine and A- and B-type motor neurons (left), alongside pHluorin imaging to visualize the extracellular pH of motor neurons (right). pHluorin, linked to the extracellular side of the mouse CD8 transmembrane domain, was expressed in cholinergic VNC motor neurons using *Punc-17Δ1*. **b** pHluorin fluorescence in motor neurons of wild type (WT) fed on the indicated bacteria. Left, fluorescence of VA5 as a representative. Solid lines and shaded regions indicate the mean and SEM, respectively. Right, comparisons of averaged pHluorin fluorescence. AU, arbitrary units. $p = 0.0006, 1, 0.00163, 0.87672, 0.0065, 0.99232, 0.00231, 0.68103, 0.00575, 0.9493, 0.01056, 1, <0.0001, 0.99373, <0.0001, 0.9137, 0.01707, 0.53622, 0.0021,$ and 1. **c** pHluorin fluorescence in motor neurons of WT, *nhx-6(ok609)*, and *nhx-6* rescue worms fed on *S. aureus*. Left, fluorescence of VA5 as a representative. Right, comparisons. Rescue was done by expressing wild-type *nhx-6* in IECs using *Pges-1*. $p = 0.0002, 0.99972, 0.0001, 0.64922, 0.00156, 0.98996, 0.0004, 0.75136, 0.00404,$ and 0.91399. **d** Spontaneous GCaMP6 signals in IECs of WT fed on the indicated bacteria. Left, sample traces. Right, comparisons of the frequency, amplitude, and inter-peak

interval of spontaneous Ca²⁺ transients. $p = 0.0009, 0.91544, 0.03181, 0.60492, 0, 0.68086, <0.0001, 0.73708, 0.00119, 0.94667, 0.02374,$ and 0.81071. **e** Spontaneous GCaMP6 signals in IECs of the indicated genotypes fed on *S. aureus*. Left, sample traces. Right, comparisons. Rescue was done by expressing wild-type *gon-2* in IECs using *Pges-1*. IEC-specific RNAi of *gtl-1* was done using *Pges-1*. $p = 0.99215, 0.99999, <0.0001, 0.91035, <0.0001, <0.0001, 1, 0.9675, <0.0001, 0.99876, <0.0001, <0.0001, 1, 0.99716, <0.0001, 1, <0.0001,$ and <0.0001. **f, g** pHluorin fluorescence in motor neurons of the indicated genotypes fed on *S. aureus*. Left, fluorescence of VA5 as a representative. Right, comparisons. IEC-specific RNAi of *cmd-1* was done using *Pges-1*. $p = 0.00622, 0.99987, <0.0001, 0.01854, 0.0003, 0.94954, <0.0001, <0.0001, 0.12486, 0.77839, 0.0003, 0.04649, 0.0001, 0.99998, <0.0001, 0.0003, 0.02043, 0.9994, 0.00111,$ and 0.00599 (**f**), and $<0.0001, <0.0001, <0.0001, <0.0001, 0.0002, 0.0005, <0.0001, 0.0002, <0.0001, <0.0001, <0.0001, <0.0001, 0.0003, 0.0008,$ and 0.00816 (**g**). *$p < 0.05$, **$p < 0.01$, and ***$p < 0.001$ (one-way ANOVA with Tukey's post hoc test). Brackets contain the number of animals tested (*n*). Data are shown as means ± SEM. Source data are provided as a Source data file.

---

and IEC-specific *gtl-1* RNAi worms responded similarly to wild type (Fig. 2e and Supplementary Fig. 3c). These results demonstrate that GON-2 is required for pathogen-enhanced Ca²⁺ influx in IECs.

We next investigated how GON-2 is activated in IECs. In wild-type fed *E. coli*, either physical immobilization using glue or pharmacological paralysis by levamisole significantly suppressed spontaneous Ca²⁺ activity, as reflected by reduced transient frequency and prolonged inter-peak intervals (Supplementary Fig. 3d). *gon-2(q362)* mutants exhibited similarly reduced Ca²⁺ activity, consistent with previous findings[60], and this defect was rescued by IEC-specific *gon-2* expression (Supplementary Fig. 3d). Notably, immobilization did not further suppress Ca²⁺ activity in *gon-2(q362)* mutants (Supplementary Fig. 3d), suggesting that GON-2 mediates mechanosensitive activation of IECs during body bending. Given that both *S. aureus* and *P. aeruginosa* damage IEC membranes and cause intestinal distention[61], we asked whether pathogen infections enhance GON-2-dependent mechanosensation. Indeed, glue immobilization suppressed pathogen-induced Ca²⁺ transients in wild type to levels observed in *gon-2(q362)* mutants, and immobilizing infected *gon-2(q362)* mutants produced no further suppression (Fig. 2e and Supplementary Fig. 3c). In *gon-2(q362)* mutants, pathogen exposure failed to alter spontaneous Ca²⁺ activity (Supplementary Fig. 3e). However, artificially inducing intestinal bloating in *E. coli*-fed worms via *aex-5* or *nhx-2* RNAi[19] failed to increase Ca²⁺ transients (Supplementary Fig. 3f), suggesting that gross tissue distention alone is insufficient to enhance GON-2 activation. Thus, IECs are mechanosensitive in a GON-2-dependent manner, and pathogen infections enhance this mechanosensory activation, likely through membrane damage that perturbs membrane tension.

Consistently, pHluorin fluorescence in VNC motor neurons was significantly increased in *gon-2(q362)* mutants compared to wild type, and this defect was rescued by IEC-specific *gon-2* expression (Fig. 2f and Supplementary Fig. 3g). pHluorin fluorescence in *nhx-6;gon-2* double mutants was indistinguishable from that of *nhx-6(ok609)* single mutants (Fig. 2f and Supplementary Fig. 3g), suggesting that GON-2 is required for NHX-6-mediated proton secretion. The calmodulin ortholog *cmd-1*[62] was also required, as IEC-specific *cmd-1* RNAi significantly increased pHluorin fluorescence in VNC motor neurons during pathogen infections, phenocopying *nhx-6(ok609)* mutants (Fig. 2g and Supplementary Fig. 3h). In *nhx-6(ok609)* mutants, IEC-specific *cmd-1* RNAi had no additional effect (Fig. 2g and Supplementary Fig. 3h). Furthermore, in *nhx-6(ok609)*, *gon-2(q362)*, *nhx-6;gon-2*, *cmd-1* RNAi, and *nhx-6;cmd-1* RNAi worms, pathogen infections failed to alter pHluorin fluorescence in VNC motor neurons (Supplementary Fig. 4e). mStrawberry fluorescence was unaffected by pathogen exposure across these genotypes (Supplementary Fig. 4c, d). Together, these results demonstrate that pathogen-induced proton

secretion from IECs is driven by mechanosensitive Ca²⁺ influx through GON-2 and is mediated by CMD-1/calmodulin-dependent activation of NHX-6.

## Intestinal pathogens activate motor neurons via NHX-6-H⁺-ASIC-1 signaling

To investigate the physiological impact of extracellular acidification on motor neurons, we recorded their spontaneous activity using GCaMP6 in worms fed on different bacteria. Compared to *E. coli*, both *S. aureus* and *P. aeruginosa* significantly increased the basal GCaMP6 fluorescence in motor neurons and the amplitude of spontaneous Ca²⁺ transients, without affecting their frequency (Fig. 3a, b). In contrast, heat-killed *S. aureus* or *P. aeruginosa-gacA(-)* had no such effect (Fig. 3a, b). Notably, these pathogen-induced effects were markedly diminished in *nhx-6(ok609)* mutants and fully rescued by expressing wild-type *nhx-6* specifically in IECs using *Pges-1* (Fig. 3c, d and Supplementary Fig. 5a, b). Similarly, *gon-2(q362)* mutants exhibited reduced Ca²⁺ activity in motor neurons, which was rescued by IEC-specific *gon-2* expression. Moreover, the *gon-2(q362)* mutation did not further reduce Ca²⁺ activity in *nhx-6(ok609)* mutants (Supplementary Fig. 5c–f), consistent with GON-2 and NHX-6 functioning in the same pathway. These results suggest that intestinal pathogens enhance the spontaneous activity of cholinergic motor neurons through a GON-2-NHX-6 signaling axis in IECs.

We next tested whether this enhanced motor neuron activity in response to pathogens requires ASIC-1. As expected, the pathogen-induced effects were significantly reduced in *asic-1(ok415)* mutants, and rescued by expressing wild-type *asic-1* in VA and VB using *Pdel-1* (Fig. 3c, d and Supplementary Fig. 5a, b), supporting the conclusion that ASIC-1 functions in VA/VB and activates DA/DB non-autonomously. Furthermore, *asic-1;nhx-6* double mutants exhibited decreased GCaMP6 signals comparable to either single mutant (Fig. 3c, d and Supplementary Fig. 5a, b). These results suggest that intestinal pathogens regulate motor neuron activity though NHX-6-H⁺-ASIC-1 signaling.

Supporting this signaling axis, NHX-6 in IECs and ASIC-1 in VA/VB are anatomically adjacent (Fig. 3e), positioning VA/VB to directly sense protons released from IECs via paracrine signaling. NHX-6 was expressed throughout the intestine and localized to both the basolateral and apical membranes of IECs, while ASIC-1 was localized to VA/VB somas and projected along their commissures (Fig. 3e), consistent with previous reports[39,52]. This spatial organization provides a structural basis for pathogen-induced motor neuron activation via NHX-6-H⁺-ASIC-1 signaling.

To determine whether ASIC-1 functions as an acid-sensor, rather than a mechanoreceptor, in activating motor neurons

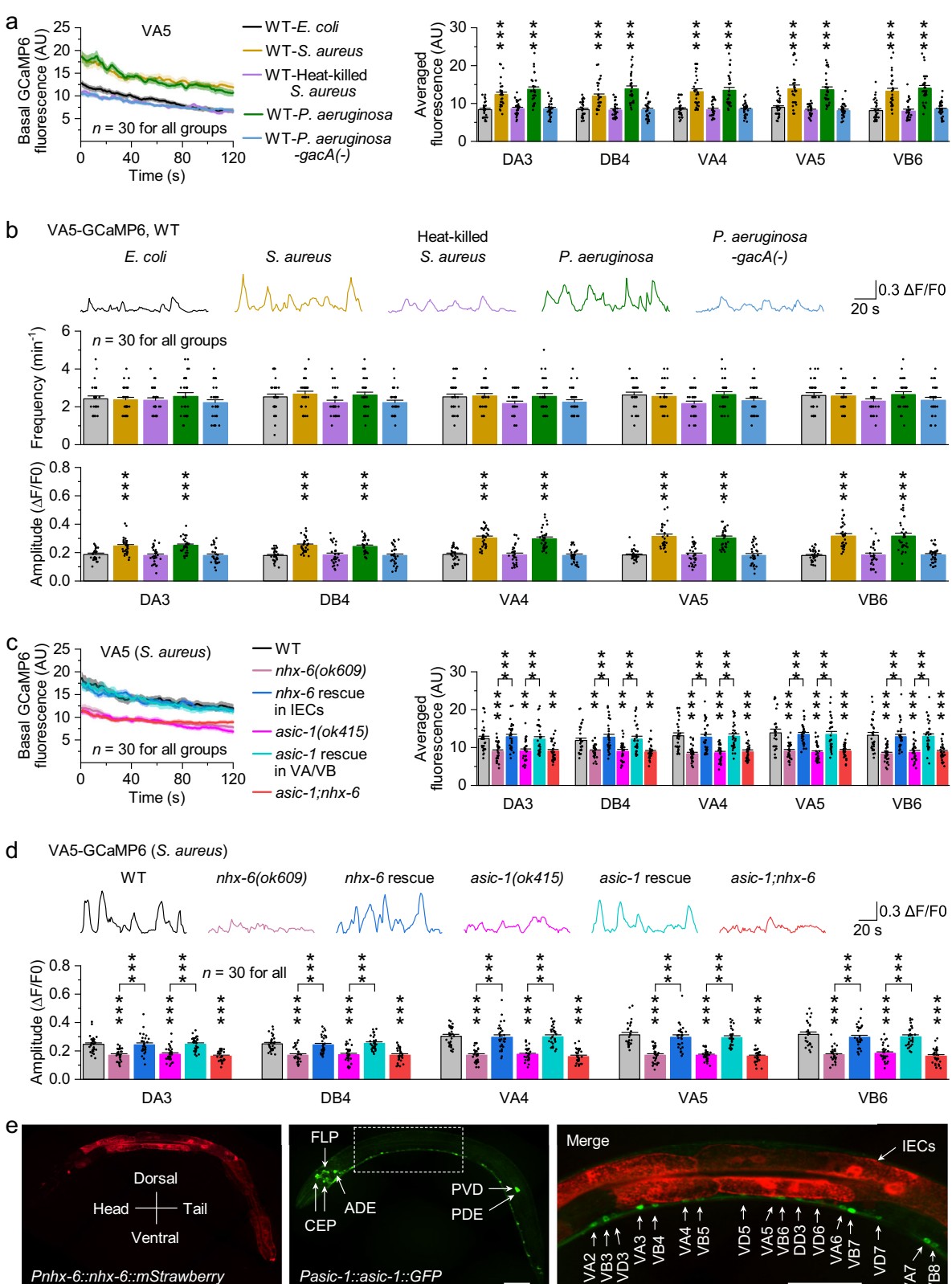

during pathogen infections, we eliminated body bending-induced mechanical stimuli by fully paralyzing worms expressing HisCl1 in body-wall muscles using 50 mM histamine[39]. To ensure sufficient bacterial ingestion despite immobility, worms were raised in liquid cultures. Under these conditions, *S. aureus* was still able to activate motor neurons in wild type, as evidenced by increased basal GCaMP6 fluorescence and spontaneous Ca$^{2+}$ amplitude,

both of which were substantially diminished in *asic-1(ok415)* mutants (Supplementary Fig. 6a, b). In contrast, deletion of *del-1*, which encodes another mechanoreceptor in VA and VB[39], did not affect *S. aureus*-induced motor neuron activation (Supplementary Fig. 6a, b). As controls, neither histamine treatment alone nor HisCl1 expression altered motor neuron activity (Supplementary Fig. 6c, d). These results support ASIC-1's role as an

**Fig. 3 | Intestinal pathogens activate motor neurons via NHX-6-H⁺-ASIC-1 signaling. a** Basal GCaMP6 fluorescence in motor neurons of wild type (WT) fed on the indicated bacteria. Left, fluorescence of VA5 as a representative. Solid lines and shaded regions indicate the mean and SEM, respectively. Right, comparisons of averaged basal fluorescence. AU, arbitrary units. $p$ = <0.0001, 0.98991, <0.0001, 1, <0.0001, 0.99999, <0.0001, 1, <0.0001, 0.9999, <0.0001, 1, <0.0001, 0.87736, <0.0001, 0.85526, <0.0001, 0.99992, <0.0001, and 0.99644. **b** Spontaneous GCaMP6 signals in motor neurons of WT fed on the indicated bacteria. Top, sample traces of VA5 as representatives. Bottom, comparisons of the frequency and amplitude of spontaneous Ca²⁺ transients. $p$ = 0.9988, 0.99684, 0.97952, 0.8736, 0.92577, 0.5859, 0.98829, 0.63829, 0.99894, 0.35548, 0.99999, 0.57157, 0.99704, 0.1462, 0.99994, 0.52712, 0.99978, 0.51004, 0.99892, and 0.67924 (top), and 0.0007, 0.98621, 0.0003, 0.97387, <0.0001, 0.99955, 0.0002, 0.99992, <0.0001, 0.99987, <0.0001, 0.99049, <0.0001, 0.99999, <0.0001, 0.99988, <0.0001, 0.99999, <0.0001, and 0.99998 (bottom). **c** Basal GCaMP6 fluorescence in motor neurons of the indicated genotypes fed on *S. aureus*. Top, fluorescence of VA5 as a representative. Bottom, comparisons. Rescues were done by expressing wild-type *nhx-6* in IECs using *Pges-1* and *asic-1* in VA/VB using *Pdel-1*, respectively. $p$ = 0.0007, 0.98201, <0.0001, 0.00123, 0.99998, 0.00236, 0.00108, 0.0092, 0.88198, 0.0002, 0.0048, 0.9971, 0.0009, 0.00288, <0.0001, 0.99715, <0.0001, <0.0001, 0.99985,

<0.0001, <0.0001, <0.0001, 0.99719, <0.0001, <0.0001, 0.99485, <0.0001, <0.0001, <0.0001, 0.99647, <0.0001, <0.0001, 0.99889, <0.0001, and <0.0001. **d** Spontaneous GCaMP6 signals in motor neurons of the indicated genotypes fed on *S. aureus*. Top, sample traces of VA5 as representatives. Bottom, comparisons. $p$ = <0.0001, 0.99996, <0.0001, <0.0001, 0.99999, <0.0001, <0.0001, <0.0001, 0.99756, <0.0001, <0.0001, 0.99967, <0.0001, <0.0001, 0, 0.99966, 0, 0, 1, 0, 0, <0.0001, 0.91779, <0.0001, <0.0001, 0.89051, 0, <0.0001, 0, 0.83476, <0.0001, 0, 0.9378, <0.0001, and 0. **e** Subcellular localization of NHX-6::mStrawberry and ASIC-1::GFP translational fusion proteins. Left, NHX-6::mStrawberry localized to the basolateral and apical membranes of IECs. Middle, ASIC-1::GFP localized to VA/VB somas and projected along their commissures (with additional expression in CEP, FLP, ADE, PVD, PDE, GABAergic motor neurons, and four unidentified tail neurons). Right, merged image of the dashed outline region showing NHX-6 in IECs and ASIC-1 in VA/VB are anatomically adjacent. A transgenic strain co-expressing NHX-6::mStrawberry and ASIC-1::GFP under *Pnhx-6* and *Pasic-1*, respectively, in a *glo-4(ok623)* mutant background was used to reduce autofluorescence. Scale bar, 50 μm. A similar pattern was observed in 18 animals. **$p$ < 0.01 and ***$p$ < 0.001 (one-way ANOVA with Tukey's post hoc test). $n$ represents the number of animals tested. Data are shown as means ± SEM. Source data are provided as a Source data file.

acid-sensor that activates motor neurons in response to pathogen infections.

## NHX-6-H⁺-ASIC-1 signaling promotes pathogen avoidance

During pathogen infections, *C. elegans* actively evades infectious bacteria to minimize exposure and enhance survival[19,20,24,25]. A- and B-type motor neurons are critical for generating the forward and backward locomotion underlying this behavior, with DA and DB innervating dorsal muscles and VA and VB innervating ventral muscles[63,64]. Given the pathogen-induced activation of motor neurons, we wondered whether NHX-6-H⁺-ASIC-1 signaling could promote pathogen avoidance by enhancing locomotion. To test this, we conducted a series of behavioral assays.

First, we quantified the locomotion speeds of wild-type worms fed different bacteria. Compared to *E. coli*, both *S. aureus* and *P. aeruginosa* significantly increased average speed, forward speed, and backward speed (Fig. 4a). In contrast, this pathogen-induced increase in locomotion speed was abolished in *asic-1(ok415)*, *nhx-6(ok609)*, and *asic-1;nhx-6* mutants, in which worms raised on pathogens exhibited locomotion speeds indistinguishable from those raised on *E. coli* (Fig. 4b and Supplementary Fig. 7a). Expressing wild-type *asic-1* in VA/VB or *nhx-6* in IECs restored pathogen-induced speed increases in the corresponding mutants (Fig. 4b and Supplementary Fig. 7a).

Second, we assessed pathogen avoidance behavior on partial lawns of bacteria. While none of wild type, *asic-1(ok415)*, *nhx-6(ok609)*, or *asic-1;nhx-6* mutants left the *E. coli* lawn within 24 hours (Supplementary Fig. 7b), they exhibited avoidance of *S. aureus* and *P. aeruginosa* shortly after ingestion (Fig. 4c and Supplementary Fig. 7c). Notably, both *asic-1(ok415)* and *nhx-6(ok609)* mutants showed significantly reduced pathogen avoidance compared to wild type, which was rescued by expressing the corresponding wild-type genes in VA/VB and IECs, respectively (Fig. 4c and Supplementary Fig. 7c). *asic-1;nhx-6* double mutants again resembled either single mutant (Fig. 4d and Supplementary Fig. 7d). Similarly, both *gon-2(q362)* mutants and worms with IEC-specific *cmd-1* RNAi displayed reduced avoidance, which was rescued in *gon-2(q362)* mutants by IEC-specific *gon-2* expression (Fig. 4e, f and Supplementary Fig. 7e, f). Notably, the *asic-1(ok415)* mutation did not further exacerbate the avoidance defects of *gon-2(q362)* mutants or *cmd-1* RNAi worms (Fig. 4e, f and Supplementary Fig. 7e, f), consistent with the role of GON-2 and CMD-1 in activating NHX-6-H⁺-ASIC-1 signaling.

Third, we tested whether exogenous H⁺ could bypass the requirement for NHX-6. H⁺ was supplemented to nematode growth medium (NGM) plates by adjusting pH to a range of 7–4. In *nhx-*

*6(ok609)* mutants, acidic pH enhanced pathogen avoidance, reaching wild-type levels at pH 4 (Supplementary Fig. 7g), indicating that extracellular H⁺ is sufficient to trigger avoidance in the absence of NHX-6. In contrast, *asic-1(ok415)* mutants failed to respond to pH 4 (Fig. 4g), demonstrating the necessity of ASIC-1 for H⁺ function. Although altering the growth medium pH is an indirect manipulation, these findings are consistent with ASIC-1-dependent proton sensing.

Finally, we expressed the mammalian capsaicin-gated cation channel TRPV1[65] in VA/VB of *asic-1(ok415)* mutants and exposed the animals to *S. aureus* in the presence of varying concentrations of capsaicin. Capsaicin had no effect on avoidance behavior in wild-type or in *asic-1* mutants, and expression of TRPV1 in VA/VB in the absence of capsaicin did not alter avoidance. In contrast, capsaicin significantly increased pathogen avoidance in the transgenic mutants in a concentration-dependent manner (Supplementary Fig. 7h), suggesting that direct activation of VA/VB is sufficient to promote avoidance behavior in the absence of NHX-6-H⁺-ASIC-1 signaling.

Together, these results demonstrate that NHX-6-H⁺-ASIC-1 signaling between the intestine and motor neurons is both necessary and sufficient for efficient pathogen avoidance during intestinal infections.

## NHX-6-H⁺-ASIC-1 signaling enhances intestinal innate immunity via cholinergic activation of the mAChR GAR2;GAR-3 in IECs

Innate immune responses in *C. elegans* IECs trigger antimicrobial production to combat ingested pathogens[18,21,22]. Previous studies have shown that neuronally released ACh enhances host resistance to *S. aureus* and *P. aeruginosa*, but the neuronal source of this cholinergic signal remains unknown[21,22]. This prompted us to investigate whether NHX-6-H⁺-ASIC-1 signaling promotes intestinal immunity by activating cholinergic VA/VB motor neurons.

To eliminate confounding effects of avoidance behavior, we performed full-lawn survival assays[32]. Both *asic-1(ok415)* and *nhx-6(ok609)* mutants exhibited significantly reduced survival during *S. aureus* or *P. aeruginosa* infection, and *asic-1;nhx-6* double mutants phenocopied the single mutants (Fig. 5a and Supplementary Fig. 8a). These defects were fully rescued by expressing wild-type *nhx-6* in IECs and *asic-1* in VA/VB, but not by expressing *asic-1* in dopaminergic neurons using *Pdat-1*[39], where ASIC-1 is also expressed[37,39] (Fig. 5a and Supplementary Fig. 8a, b), supporting a gut-to-motor neuron axis. Furthermore, *nhx-6(ok609)*, but not *asic-1(ok415)* mutants, exhibited improved survival on pH 4 NGM plates during infections (Fig. 5b and Supplementary Fig. 8c), confirming a pH-dependent activation of ASIC-1 downstream of NHX-6. These results suggest that NHX-6-H⁺-ASIC-1 signaling across IECs and VA/VB promotes host defense.

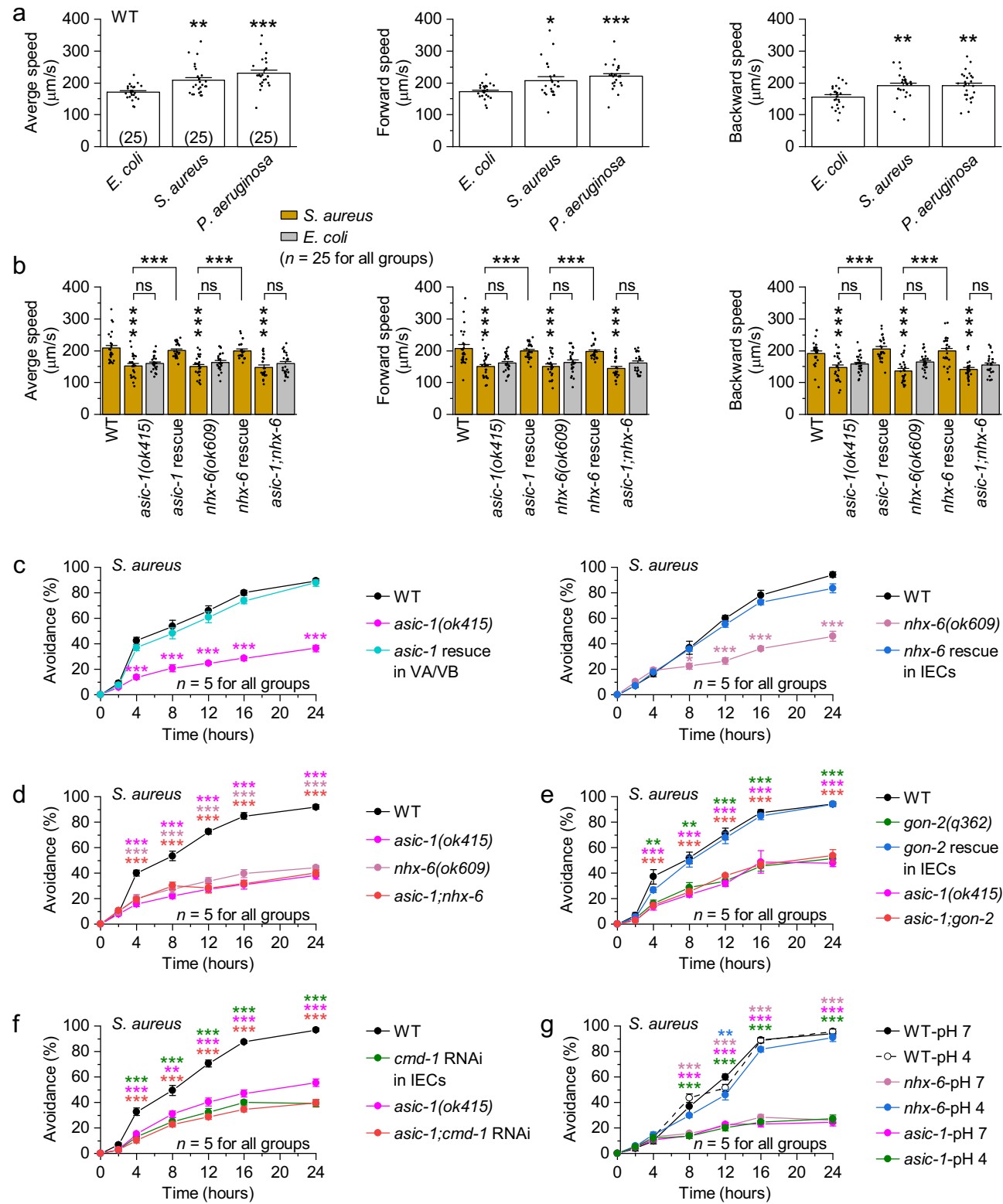

Consistent with previous findings[21,22], deletion of *cwn-2* (Wnt ligand) or *pmk-1* (p38 MAPK) reduced survival during pathogen infections (Fig. 5c and Supplementary Fig. 8d). These defects were greatly rescued by IEC-specific expression and recapitulated by IEC-specific RNAi (Fig. 5c and Supplementary Fig. 8d). *asic-1;cwn-2* and *asic-1;pmk-1* double mutants resembled *cwn-2(ok895)* and *pmk-1(km25)* single mutants, respectively (Fig. 5c and Supplementary Fig. 8d). During *S. aureus* infection, both *asic-1(ok415)* and *nhx-6(ok609)* mutants showed reduced mRNA levels of cwn-2 and five Wnt-

dependent immune genes (*ilys-2, ilys-3, lys-5, cpr-2,* and *clec-60*)[21], as well as decreased expression of the *Pclec-60::GFP* reporter[21] (Fig. 5d, e). Similarly, during *P. aeruginosa* infection, both mutants exhibited reduced PMK-1 phosphorylation, lower mRNA levels of six PMK-1-dependent immune genes (*lys-8, K08D8.5, irg-4, sysm-1, F55G11.4,* and *irg-5*)[22], and weaker *Pirg-5::GFP* reporter[28] expression (Supplementary Fig. 8e–g). The defective immune reporter expression was rescued by expressing wild-type *nhx-6* in IECs and *asic-1* in VA/VB, respectively (Fig. 5e and Supplementary Fig. 8f). *asic-1;nhx-6* double mutants

**Fig. 4 | NHX-6-H$^+$-ASIC-1 signaling promotes pathogen avoidance by increasing locomotion speed. a** Comparisons of average, forward, and backward speed among wild type (WT) fed on *E. coli*, *S. aureus*, or *P. aeruginosa*. $p = 0.0048$, <0.0001, 0.01297, 0.0003, 0.00622, and 0.00735. **b** Comparisons of average, speed, and backward speed among the indicated genotypes fed on *S. aureus*. Rescues were done by expressing wild-type *nhx-6* in IECs using *Pges-1* and *asic-1* in VA/VB using *Pdel-1*, respectively. $p$ = <0.0001, 0.99584, 0.99843, <0.0001, <0.0001, 0.89361, 0.99114, <0.0001, <0.0001, 0.93998, <0.0001, 0.97732, 0.99418, <0.0001, <0.0001, 0.95377, 0.98715, 0.0002, <0.0001, 0.71338, 0.0006, 0.97444, 0.90092, <0.0001, <0.0001, 0.16692, 0.99852, <0.0001, <0.0001, and 0.86946.
**c–f** Pathogen avoidance of the indicated genotypes on the partial lawn of *S. aureus*. $p = 0.42454$, <0.0001, 0.0002, <0.0001, <0.0001, <0.0001, 0.90029, 0.34903, 0.71227, 0.67958, 0.13478, 0.9502, 0.42248, 0.63548, 0.030617, <0.0001, <0.0001, <0.0001, 0.99799, 0.93486, 0.9916, 0.53257, 0.44255, and 0.11867 (**c**), 0.9995, <0.0001, <0.0001, <0.0001, <0.0001, <0.0001, 0.99966, <0.0001, <0.0001,

<0.0001, <0.0001, <0.0001, 0.98977, <0.0001, 0.0002, <0.0001, <0.0001, and <0.0001 (**d**), 0.99857, 0.00194, 0.00141, <0.0001, <0.0001, <0.0001, 0.99998, 0.29021, 0.99974, 0.99952, 0.99997, 1, 0.84131, 0.00059, <0.0001, <0.0001, 0.0001, <0.0001, 0.79112, 0.00083, 0.00031, <0.0001, <0.0001, and <0.0001 (**e**), and 0.5046, <0.0001, 0.00018, <0.0001, <0.0001, <0.0001, 0.54465, 0.00018, 0.00263, <0.0001, <0.0001, <0.0001, 0.53034, <0.0001, <0.0001, <0.0001, <0.0001, and <0.0001 (**f**). **g** Pathogen avoidance of WT, *nhx-6(ok609)*, and *asic-1(ok415)* worms on the partial lawn of *S. aureus* at pH 7 or 4. $p = 1$, 0.99908, 0.88303, 0.31931, 1, 1, 1, 0.99935, 0.0009, <0.0001, 0, <0.0001, 0.99983, 0.93851, 0.84253, 0.00821, 0.1449, 0.9994, 1, 1, 0.00028, <0.0001, 0, <0.0001, 1, 1, 0.0003, <0.0001, 0, and <0.0001. *$p < 0.05$, **$p < 0.01$, and ***$p < 0.001$ (one-way ANOVA with Tukey's post hoc test for (**a**, **b**); two-way repeated-measures ANOVA with Sidak's multiple-comparisons correction for (**c–g**). ns, no significance. Brackets contain the number of animals tested (**a**, **b**). *n* represents the number of independent assays (**c–g**). Data are shown as means ± SEM. Source data are provided as a Source data file.

phenocopied the single mutants (Fig. 5d, e and Supplementary Fig. 8e–g). Notably, the immune defects of *nhx-6(ok609)*, but not *asic-1(ok415)*, were rescued by pH 4 (Fig. 5f, g and Supplementary Fig. 8h–j), further supporting the role of proton-mediated ASIC-1 activation in host defense. These results suggest that NHX-6-H$^+$-ASIC-1 signaling enhances intestinal immunity through the Wnt and PMK-1/p38 MAPK pathways.

Notably, *asic-1(ok415)*, *nhx-6(ok609)*, and *asic-1;nhx-6* double mutants showed normal pharyngeal pumping rates, defecation cycle durations, and intestinal fluorescence of ingested *E. coli* OP50-GFP in 24 hours (Supplementary Fig. 9a–c). However, these mutants accumulated significantly more *S. aureus* in the intestine, as indicated by increased colony-forming units (CFUs) at 16 and 24 hours (Supplementary Fig. 9d). These results suggest that while the mutants can normally ingest, process, and expel bacteria, they are defective in eliminating ingested pathogens, reinforcing the immune-promoting role of NHX-6-H$^+$-ASIC-1 signaling.

In cholinergic neurons, ACh is synthesized by the choline acetyltransferase CHA-1 and loaded into synaptic vesicles by the transporter UNC-17[66]. We found that VA/VB-targeted *cha-1* RNAi worms and *unc-17(e245)* mutants both showed reduced survival during pathogen infections (Fig. 5h and Supplementary Fig. 10a). The defect of *unc-17(e245)* mutants was greatly rescued by expressing wild-type *unc-17* in VA/VB and phenocopied by VA/VB-targeted *unc-17* RNAi (Fig. 5h and Supplementary Fig. 10a). Importantly, *asic-1(ok415)* did not further reduce survival in VA/VB-targeted *unc-17* RNAi worms (Fig. 5h and Supplementary Fig. 10a). VA/VB-targeted *unc-17* RNAi did not further decrease survival in *cwn-2(ok895)* or *pmk-1(km25)* mutants (Fig. 5h and Supplementary Fig. 10a). During pathogen-infections, *unc-17(e245)* mutants exhibited reduced mRNA levels of *cwn-2*, Wnt- and PMK-1-dependent immune genes, and decreased PMK-1 phosphorylation (Fig. 5i and Supplementary Fig. 10b, c). VA/VB-targeted *unc-17* RNAi did not further suppress immune reporter expression in *asic-1(ok415)* mutants (Fig. 5j and Supplementary Fig. 10d). These results suggest that ACh released from VA/VB acts upstream of Wnt and PMK-1/p38 MAPK signaling to mediate the immune-promoting effects of NHX-6-H$^+$-ASIC-1 signaling.

It was previously demonstrated that ACh induced by *S. aureus* infection activates the mAChR GAR-2;GAR-3 in IECs, triggering canonical Wnt signaling and antimicrobial gene expression[21]. To determine whether *P. aeruginosa*-induced ACh release also signals through GAR-2;GAR-3 to activate PMK-1, we analyzed *gar-2(ok520);gar-3(gk305)* double mutants. These mutants exhibited decreased survival during both *S. aureus* and *P. aeruginosa* infections, which was rescued by IEC-specific expression of wild-type genes and recapitulated by IEC-specific RNAi (Supplementary Fig. 10e, f). In contrast, deletion of *gar-1*, encoding another mAChR, did not affect survival during pathogen

infections (Supplementary Fig. 10g). During *P. aeruginosa* infection, *gar-2;gar-3;pmk-1* triple mutants showed similarly reduced survival to *pmk-1(km25)* single mutants (Supplementary Fig. 10h). *gar-2;gar-3* mutants also exhibited reduced PMK-1 phosphorylation, lower mRNA levels of PMK-1-dependent immune genes, and diminished immune reporter expression (Supplementary Fig. 10i–k). Furthermore, VA/VB-targeted *unc-17* RNAi did not further reduce immune reporter expression and survival in *gar-2;gar-3* mutants (Supplementary Fig. 10k, l). Pharmacological treatment with oxotremorine, a muscarinic agonist[21], significantly increased PMK-1 phosphorylation and improved survival in *unc-17(e245)* mutants during *P. aeruginosa* infection (Supplementary Fig. 10m, n). These results suggest that GAR-2;GAR-3 acts downstream of ACh released from cholinergic motor neurons to activate both the Wnt and PMK-1/p38 MAPK pathways in IECs during pathogen infections.

Finally, *asic-1;gar-2;gar-3* triple mutants exhibited reduced survival and immune reporter expression similar to *gar-2;gar-3* double mutants during pathogen infections (Fig. 5k, l and Supplementary Fig. 10o, p). Together, these results suggest that NHX-6-H$^+$-ASIC-1 signaling enhances intestinal immunity by promoting cholinergic activation of GAR-2;GAR-3 in IECs.

## Mouse ASIC1a and NHE1 can functionally substitute for their counterparts in *C. elegans*

*C. elegans* ASIC-1 shares extensive sequence and overall structure similarities with mammalian ASIC1a, which forms proton-gated homomeric channels expressed in peripheral sensory neurons, including vagal afferents[38,49]. Among mammalian NHEs, NHE1 is known to localize to the basolateral membrane of IECs[67].

To test whether the NHX-6-ASIC-1 signaling axis is functionally conserved, we ectopically expressed mouse ASIC1a (mASIC1a) in VA/VB and mouse NHE1 (mNHE1) in IECs of *C. elegans*, respectively. In *asic-1(ok415)* mutants, pH 4-evoked calcium responses in A- and B-type motor neurons were rescued by mASIC1a expression (Fig. 6a and Supplementary Fig. 11a). In *nhx-6(ok609)* mutants during pathogen infections, expression of mNHE1 normalized pHluorin fluorescence in motor neurons to wild-type levels (Fig. 6b and Supplementary Fig. 11b). mStrawberry fluorescence was unaffected across these rescue conditions (Supplementary Fig. 11c). Additionally, spontaneous motor neuron activity was rescued by expressing mASIC1a in *asic-1(ok415)* or mNHE1 in *nhx-6(ok609)* (Fig. 6c,d and Supplementary Fig. 11d,e). Correspondingly, mutants expressing the respective mouse genes displayed wild-type levels of pathogen avoidance and survival against pathogens (Fig. 6e,f and Supplementary Fig. 11f,g). Together, these results demonstrate that mouse ASIC1a and NHE1 can functionally substitute for *C. elegans* ASIC-1 and NHX-6 in vivo, supporting the potential evolutionary conservation of this epithelial-neural signaling pathway.

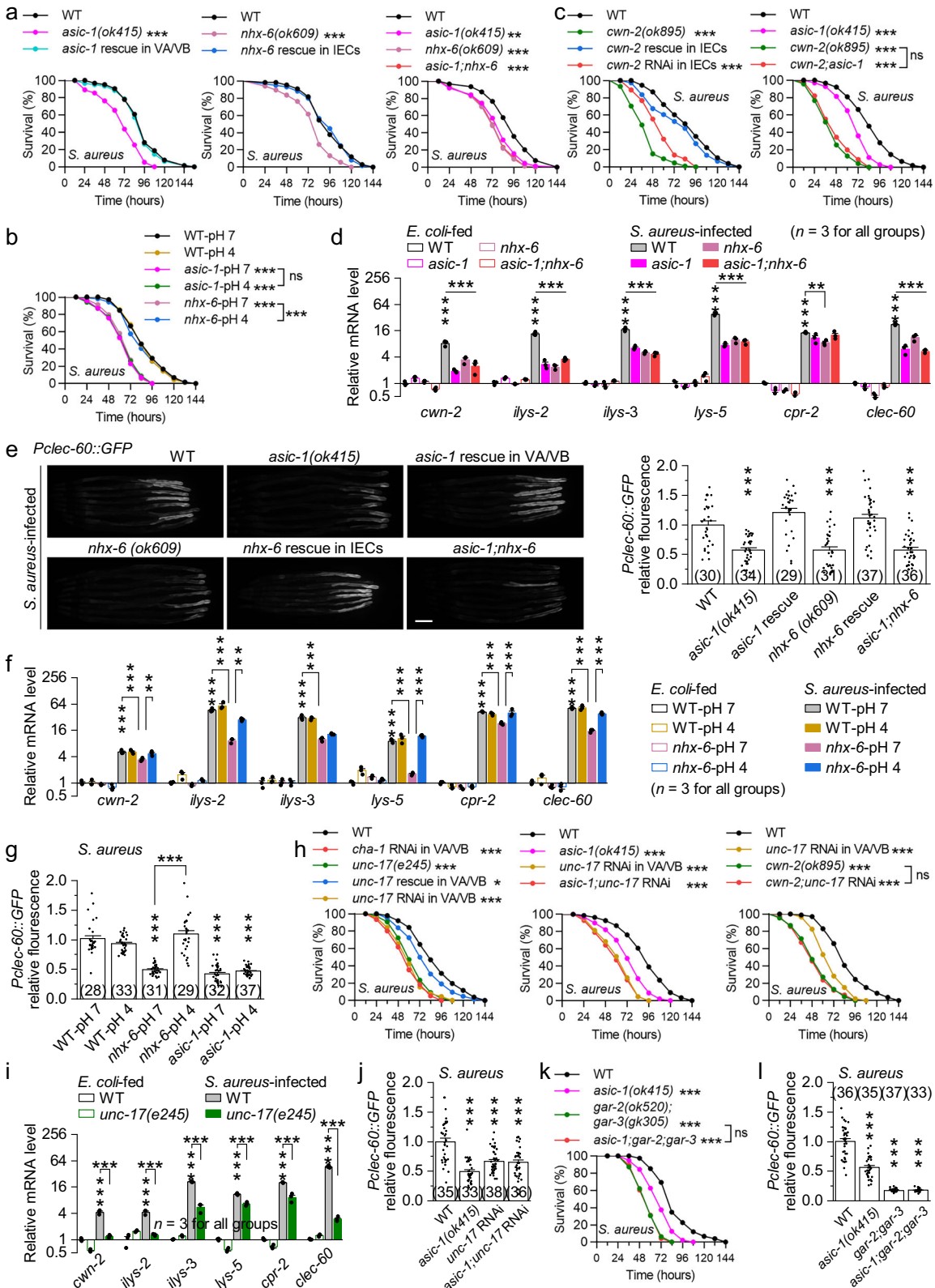

## Discussion

Coordinated host defense is essential for animal survival against pathogen attack[1,3,6]. IECs are strategically positioned to interact with ingested pathogens and relay intestinal information to the nervous system, thereby orchestrating behavioral and immune responses[9–11]. However, the molecular mechanisms that integrate intestinal pathogen detection with systemic host defenses remain poorly understood.

Here, we identify a proton-mediated gut-to-neuron signaling pathway that integrates behavioral and immune defenses in *C. elegans*. Intestinal pathogens stimulate mechanosensitive $Ca^{2+}$ influx into IECs through the TRP channel GON-2, activating NHX-6 through CMD-1 and leading to basolateral proton secretion. These protons enhance the basal activity of cholinergic VNC motor neurons via ASIC-1, facilitating ACh release. This heightened cholinergic transmission promotes both

**Fig. 5 | NHX-6-H⁺-ASIC-1 signaling enhances intestinal innate immunity via the mAChR GAR-2;GAR-3. a, c** Percent survival of the indicated genotypes on the full lawn of *S. aureus*. WT, wild type. Rescues were done by expressing wild-type *asic-1* in VA/VB using *Pdel-1*, *nhx-6* in IECs using *Pges-1*, and *cwn-2* in IECs using *Pges-1*, respectively. IEC-specific RNAi of *cwn-2* was done using *Pges-1*. *p* = <0.0001, 0.6599, <0.0001, 0.9271, 0.0011, <0.0001, and <0.0001 (**a**), and <0.0001, 0.1252, <0.0001, <0.0001, <0.0001, <0.0001, and 0.1422 (**c**). **b** Percent survival of WT, *asic-1(ok415)*, and *nhx-6(ok609)* worms on the full lawn of *S. aureus* at pH 7 or 4. *p* = 0.8904, <0.0001, <0.0001, 0.8046, <0.0001, 0.471, and <0.0001. **d** qRT-PCR analyses of the expression levels of *cwn-2* and five Wnt-dependent immune genes in the indicated genotypes fed on *E. coli* or *S. aureus*. *p* = 0.99656, 1, 0.99842, <0.0001, <0.0001, <0.0001, <0.0001, 0.99515, 1, 0.99951, 0, 0, 0, 0, 1, 1, 0.99999, 0, 0, 0, 0, 1, 1, 0.99943, 0, 0, 0, 0, 0.99999, 0.99999, 0.99991, <0.0001, 0.07395, 0.00119, 0.68578, 1, 0.99847, 1, 0, 0, <0.0001, and 0. **e** Expression of *Pclec-60::GFP* in the indicated genotypes fed on *S. aureus*. Shown are representative images and comparisons of relative fluorescence normalized to wild-type controls. Scale bars, 100 μm. *p* = <0.0001, 0.14304, <0.0001, 0.70273, and <0.0001. **f** qRT-PCR analyses of the expression levels of *cwn-2* and five Wnt-dependent immune genes in WT and *nhx-6(ok609)* worms fed on *E. coli* or *S. aureus* at pH 7 or 4. *p* = 1, 0.99999, 0.99585, <0.0001, 1, 0.0007, 0.88882, 0.00791, 1, 1, 1, <0.0001, 0.04082, <0.0001, 0.00153, 0.00118, 1, 1, 1, 0, 0.92138, <0.0001, <0.0001, 0.33197, 0.91704, 0.99971, 1, <0.0001, 0.67172, <0.0001, 0.05008, <0.0001, 1, 1, 1, <0.0001, 0.53153, <0.0001, 0.98187, 0.0001, 1, 1, 1, 0, 0.99925, <0.0001, 0.00118, and <0.0001. **g** Comparisons of *Pclec-60::GFP* relative fluorescence in WT, *asic-1(ok415)*, and *nhx-6(ok609)* worms fed on *S. aureus* at pH 7 or 4. *p* = 0.58136, 0, 0.6396, 0.01727, 0, 0, and 0.92422. **h, k** Percent survival of the indicated genotypes on the full lawn of *S. aureus*. Rescue and RNAi of *unc-17* were done in VA/VB using *Pdel-1*. *p* = <0.0001, <0.0001, 0.0463, <0.0001, <0.0001, <0.0001, <0.0001, <0.0001, <0.0001, <0.0001, and 0.8229 (**h**), and <0.0001, <0.0001, <0.0001, and 0.8665 (**k**). **i,** qRT-PCR analyses of the expression levels of *cwn-2* and five Wnt-dependent immune genes in WT and *unc-17(e245)* worms fed on *E. coli* or *S. aureus*. *p* = 0.2119, <0.0001, <0.0001, 0.18919, <0.0001, <0.0001, 0.9856, 0, 0, 0.58112, 0, <0.0001, 0.98226, 0, <0.0001, 0.99804, 0, and 0. **j, l** Comparisons of *Pclec-60::GFP* relative fluorescence in the indicated genotypes fed on *S. aureus*. RNAi of *unc-17* was done in VA/VB using *Pdel-1*. *p* = 0, <0.0001, and <0.0001 (**j**), and 0, 0, and <0.0001 (**l**). **p* < 0.01 and ***p* < 0.001 (log-rank (Kaplan-Meier) test for (**a–c**, **h**, **k**); one-way ANOVA with Tukey's post hoc test for (**d–g**, **i**, **j**, **l**)). ns, no significance. Brackets contain the number of animals tested (*n*). Data are shown as means ± SEM. Source data are provided as a Source data file.

pathogen avoidance behavior and intestinal innate immunity (Fig. 7). Thus, proton-mediated gut-to-neuron signaling provides an integrated mechanism for coordinating host defense against intestinal infection.

By routing epithelial sensing signals through the nervous system, the IEC-neuron-IEC loop enables rapid coupling of behavioral avoidance with intestinal immunity, while affording amplification, flexibility, and integration with internal state that cannot be achieved by cell-autonomous IEC signaling alone. Within this framework, proton-mediated activation of motor neurons enhances locomotor dynamics associated with pathogen avoidance, while remaining embedded within a broader motor circuit subject to modulation by upstream interneurons and neuromodulatory signals. In parallel, Ca²⁺/calmodulin signaling may also directly engage PMK-1/p38 MAPK or other immune signaling pathways cell-autonomously in IECs, as has been demonstrated for p38 MAPK regulation in AWC olfactory sensory neurons[68], providing an additional layer of regulation that complements gut-to-neuron communication.

Notably, mouse NHE1 and ASIC1a can functionally substitute for NHX-6 and ASIC-1 in *C. elegans*, demonstrating that the molecular architecture of this signaling axis is functionally compatible across species. This cross-species substitution underscores the potential evolutionary conservation of proton-mediated epithelial-neural communication in coordinating host defense.

Protons, long recognized as ubiquitous biological ions, are increasingly appreciated as intercellular transmitters[50,55,69]. Our study demonstrates a role for protons as gut-derived signaling molecules that convey epithelial stress signals to the nervous system, initiating protective behavioral and immune responses. Given that IECs can secrete protons under a range of physiological and pathological conditions across species[48–50,53], the broader functional relevance of proton signaling warrants further investigation. We propose that protons may act synergistically or in parallel with other intestinal signals, encoding distinct or overlapping information to coordinate protective responses at both local and systemic levels.

While our data demonstrate that pathogen infections trigger NHX-6-dependent proton release from IECs, the immediate intracellular source of these protons cannot be conclusively resolved by our experiments. IECs in *C. elegans* maintain dynamic proton pools associated with rhythmic intestinal pH oscillations, which are driven in part by proton movement across the apical membrane from the intestinal lumen[53]. Pathogen exposure may therefore promote basolateral release of pre-existing proton gradients. In addition, intracellular acidic compartments could contribute to this response. Lysosome-related organelles in IECs are disrupted by pathogen-derived toxins

such as pyocyanin from *P. aeruginosa*[70], raising the possibility that infection-associated stress alters intracellular proton handling. However, lysosomal proton efflux mechanisms remain poorly defined in *C. elegans*, and key lysosomal proton channels such as TMEM175[71] lack clear nematode orthologs. Thus, our findings define the signaling and transport machinery required for pathogen-evoked proton release, while leaving open the precise subcellular origin of the protons involved.

In *C. elegans*, ASIC-1 also functions as a mechanoreceptor in VA and VB motor neurons[39], which form commissures encircling the intestine[47]. Coincidentally, ingested pathogens cause intestinal distention[19], raising the possibility that ASIC-1 integrates both chemical and mechanical signals. Similarly, mammalian ASICs respond to both acidic pH and mechanical stimuli[49,72], and are expressed in gut-innervating sensory neurons[37,38,49], positioning them to detect IEC-derived protons and gastrointestinal distention[73]. These parallels suggest that ASICs may serve conserved dual roles as chemo- and mechanosensors in gut-to-neuron communication.

We show that NHX-6-H⁺-ASIC-1 signaling in *C. elegans* translates intestinal pathogen infection into motor neuron activation, increasing locomotion speed to facilitate pathogen avoidance. This behavioral response resembles conditioned flavor avoidance and other adaptive defensive behaviors observed in mammals[3]. Supporting this notion, ASICs are widely expressed in vagal, spinal, and enteric neurons, which transmit sensory information from the gut to the brain and regulate defensive responses, as well as intestinal motility, barrier integrity, and immune responses[4,7,11,15,49,74–76]. Similarly, NHEs localize to the basolateral membrane of intestinal epithelial cells, positioning them to regulate extracellular proton flux[67,77]. Together, these parallels suggest that epithelial-derived protons may serve as a conserved signal coupling local epithelial states to neural circuits controlling defensive and homeostatic processes.

Effective host defense requires both accurate pathogen detection and rapid activation of protective responses. In addition to canonical pathogen recognition pathways, mechanosensation has emerged as an important trigger of immune cell activation[56,78,79]. Despite lacking classical pattern recognition receptors, *C. elegans* mounts both shared and pathogen-specific immune defenses[2,21,61,80]. Although *S. aureus* and *P. aeruginosa* elicit divergent transcriptional responses[61], proton-mediated gut-to-neuron signaling likely represents a shared physiological readout of intestinal epithelial stress, while pathogen specificity is conferred by parallel signaling pathways that drive distinct transcriptional and immune programs. Consistent with this model, we find that these phylogenetically distinct pathogens enhance IEC

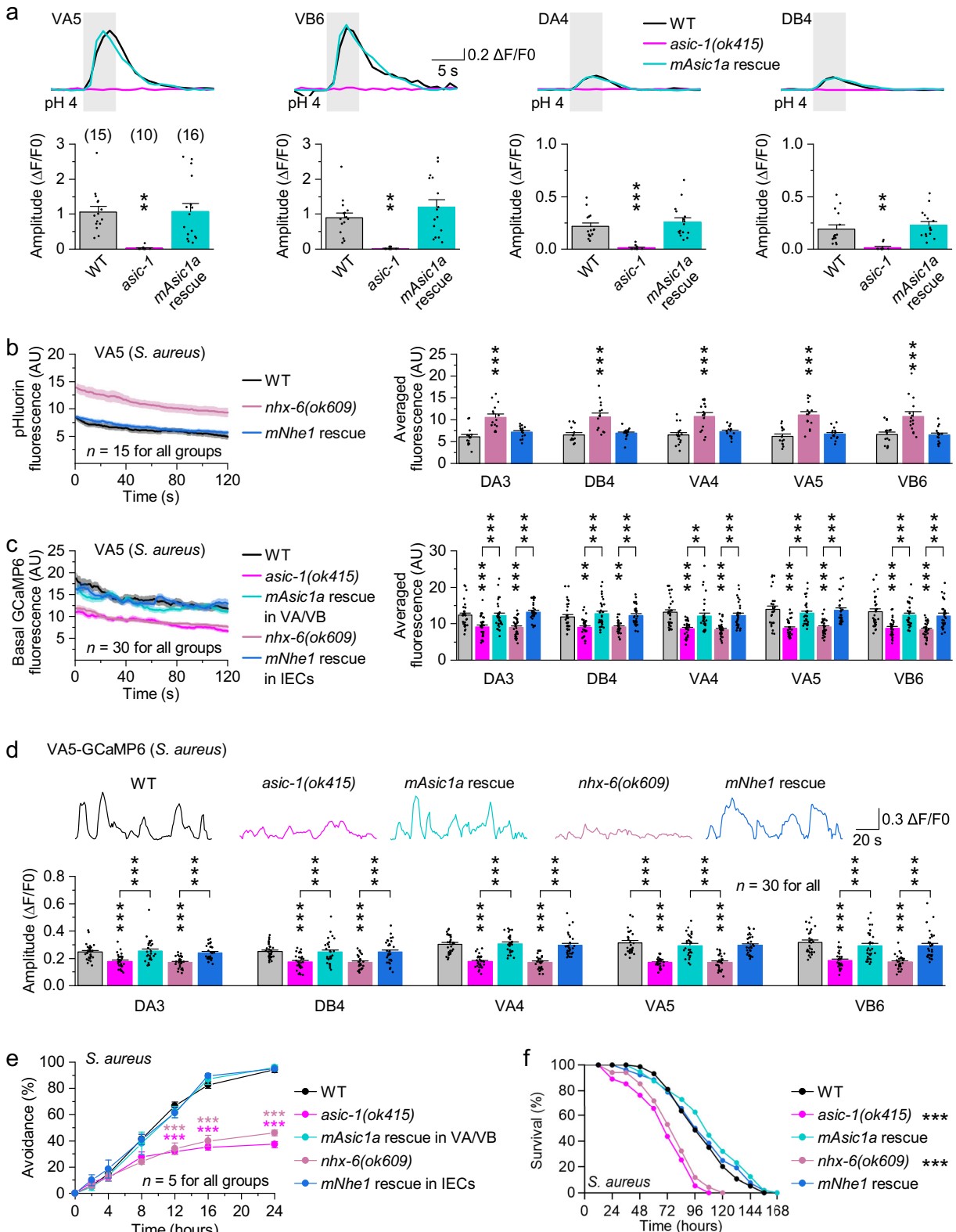

mechanosensitive Ca²⁺ influx via the TRP channel GON-2, while not excluding contributions from other sources of intracellular Ca²⁺ signaling. This infection-associated enhancement likely reflects localized epithelial stress or membrane perturbation arising during infections[57,61,81–83], operating within a spatially and temporally restricted physiological regime that precedes severe epithelial compromise. While intestinal distention can activate innate immunity[19], our data

indicate that gross distention is insufficient to drive GON-2-dependent Ca²⁺ influx, pointing instead to more localized changes in membrane tension or integrity associated with epithelial damage. These findings define a form of damage-associated immune surveillance that links epithelial sensing to neural control of host defense, functioning alongside established pathogen-sensing pathways, other neuronal inputs, and parallel epithelial immune mechanisms[2,5,16,61,70,80,84].

**Fig. 6 | Mouse ASIC1a and NHE1 can functionally substitute for their counterparts in *C. elegans*. a** pH 4-evoked calcium responses in motor neurons of wild type (WT), *asic-1(ok415)* mutants, and *mAsic1a* rescue worms. Top, sample traces. Bottom, comparisons. Rescue was done by expressing mouse *Asic1a* (*mAsic1a*) in VA and VB using *Pdel-1*. $p$ = 0.00135, 0.99793, 0.00346, 0.36052, 0.0008, 0.63501, 0.00395, and 0.69052. **b** pHluorin fluorescence in motor neurons of WT, *nhx-6(ok609)* mutants, and *mNhe1* rescue worms fed on *S. aureus*. Left, fluorescence of VA5 as a representative. Solid lines and shaded regions indicate the mean and SEM, respectively. Right, comparisons of averaged pHluorin fluorescence. AU, arbitrary units. Rescue was done by expressing mouse *Nhe1* (*mNhe1*) in IECs using *Pges-1*. $p$ = <0.0001, 0.37489, <0.0001, 0.90802, <0.0001, 0.62289, <0.0001, 0.80592, 0.0006, and 0.99773. **c** Basal GCaMP6 fluorescence in motor neurons of the indicated genotypes fed on *S. aureus*. Left, fluorescence of VA5 as a representative. Right, comparisons of averaged basal fluorescence. $p$ = 0.0004, 0.99977, 0.0008, 0.0002, 0.79371, <0.0001, 0.00222, 0.77236, <0.0001, 0.00444, 0.98847, 0.0008, <0.0001, 0.73893, 0.00112, <0.0001, 0.0004, 0.0002, <0.0001, 0.76468, <0.0001, <0.0001, 0.99829, <0.0001, <0.0001, 0.79071, 0.0002, <0.0001, 0.6768, and

<0.0001. **d** Spontaneous GCaMP6 signals in motor neurons of the indicated genotypes fed on *S. aureus*. Top, sample traces. Bottom, comparisons of the amplitude of spontaneous $Ca^{2+}$ transients. $p$ = <0.0001, 0.9804, <0.0001, <0.0001, 0.99841, <0.0001, 0.0002, 0.99925, 0.0006, 0.0003, 0.0007, 0.0008, <0.0001, 0.9986, 0, 0, 0.98779, <0.0001, 0, 0.81367, <0.0001, 0, 0.82522, <0.0001, <0.0001, 0.71405, <0.0001, <0.0001, 0.73148, and <0.0001. **e** Pathogen avoidance of the indicated genotypes on the partial lawn of *S. aureus*. $p$ = 0.99699, 0.99995, 0.47386, <0.0001, <0.0001, <0.0001, 1, 1, 1, 0.98898, 0.9855, 0.99955, 1, 1, 0.18524, <0.0001, <0.0001, <0.0001, 0.95979, 0.99951, 1, 0.97315, 0.75048, and 1. **f** Percent survival of the indicated genotypes on the full lawn of *S. aureus*. $p$ = <0.0001, 0.1744, <0.0001, and 0.7989. **$p$ < 0.01 and ***$p$ < 0.001 (one-way ANOVA with Tukey's post hoc test for (**a**–**d**); two-way repeated-measures ANOVA with Sidak's multiple-comparisons correction for (**e**); log-rank (Kaplan-Meier) test for (**f**)). Brackets contain the number of animals tested (**a**). $n$ represents the number of animals tested (**b**–**d**) or independent assays (**e**). Data are shown as means ± SEM. Source data are provided as a Source data file.

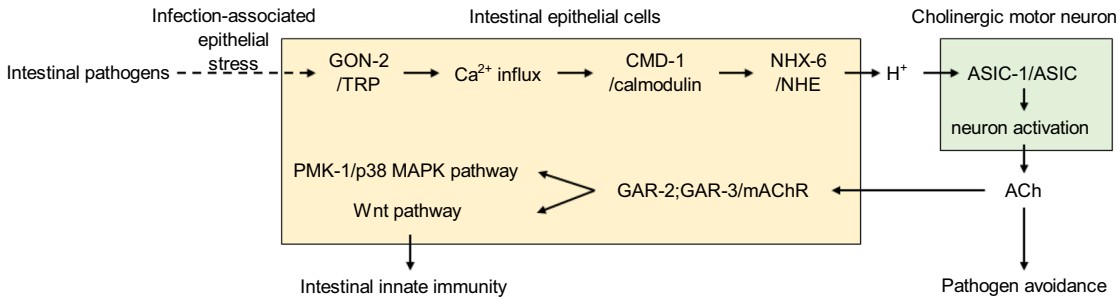

**Fig. 7 | Working model of NHX-6-H⁺-ASIC-1 signaling in coordinating behavioral and innate immune defenses against intestinal pathogens.** Intestinal pathogens enhance GON-2-dependent mechanosensitive $Ca^{2+}$ influx into IECs, likely reflecting infection-associated epithelial stress or membrane perturbation. $Ca^{2+}$-dependent activation of the $Na^+/H^+$ exchanger NHX-6 via calmodulin CMD-1 promotes basolateral proton secretion. These protons activate the acid-sensing ion channel ASIC-1 in cholinergic motor neurons, thereby enhancing cholinergic transmission to promote both pathogen avoidance and intestinal innate immunity.

However, whether GON-2 itself constitutes the mechanosensitive pore-forming channel remains an open question. GON-2 may instead function as an essential component or regulator of a mechanosensitive channel complex, or influence channel expression, assembly, or localization. Notably, the mammalian gut expresses numerous TRP channels and experiences continuous mechanical stimulation via peristalsis[85]. In innate immune cells, TRPV4 mediates tension-induced $Ca^{2+}$ influx required for immune activation[56]. Thus, pathogen-induced epithelial stress may represent a conserved danger signal that engages proton-mediated gut-to-neuron signaling to coordinate host defense.

Several neurotransmitters, including ACh, GABA, dopamine, and octopamine, have been implicated in regulating intestinal immunity in *C. elegans*[21,22,28,29]. However, how intestinal infections influence these neuronal transmissions remains largely unknown. A- and B-type motor neurons can modulate interneurons via antidromic-rectifying gap junctions[39,86] and may influence higher-order neurons through more complex circuits[41], suggesting that intestinal proton release could engage broad neural networks to coordinate immune surveillance and systemic responses. Moreover, ASICs are expressed in multiple neuronal populations[40,46], implying that proton-mediated signaling could influence multiple behavioral and physiological processes beyond pathogen avoidance.

While neural activation promotes immunity, excessive responses must be restrained to preserve homeostasis. The nervous system maintains this balance by exerting both stimulatory and inhibitory control[5,6]. *In C. elegans*, as in mammals, cholinergic signaling enhances intestinal innate immunity[21,22], paralleling cholinergic regulation of Paneth cell antimicrobial secretion[7,11–13]. Conversely, dopamine and octopamine suppress intestinal immunity[28,29], mirroring the inhibitory actions of their mammalian

analog, catecholamine, on mammalian innate responses[6,7]. These parallels underscore the conserved logic of neuroimmune coordination across species.

In summary, our study identifies a proton-mediated gut-to-neuron signaling pathway that coordinates behavioral and immune defenses against intestinal pathogens (Fig. 7). This axis is built upon conserved anatomical and molecular components: intestinal epithelial proton exporters (NHX/NHE family), acid-sensitive neuronal channels (ASICs), and cholinergic feedback mechanisms. Downstream, p38 MAPK and Wnt pathways drive IEC immunity in both species[21,26,87–89]. Given these shared features, proton-mediated gut-to-neuron signaling likely represents an evolutionarily conserved and generalizable strategy for integrating local intestinal cues with systemic host defense. Our findings provide a framework for identifying potential therapeutic targets in gastrointestinal disorders associated with pathogen infections and inflammation.

## Methods

### Bacterial strains and culture

Bacterial strains *E. coli* OP50, *S. aureus* NCTC8325, and *P. aeruginosa* PA14 were cultured under conditions optimized for their growth and virulence[90]. *E. coli* was cultured in LB medium. *P. aeruginosa* and *S. aureus* were cultured in LB and tryptic soy broth (Hopebio) containing 10 μg/ml nalidixic acid (Aladdin), respectively, with gentle shaking. All bacteria were grown overnight at 37 °C and subsequently seeded onto 3.5-cm agar plates. Specifically, *E. coli* was seeded on NGM plates and incubated at 37 °C for 20 h. *P. aeruginosa* was seeded on modified NGM plates containing 0.35% peptone (Aobox) and 2 mg/ml tryptophan (Aladdin), followed by incubation at 37 °C for 24 h and 25 °C for an additional 6 h. A 1:5 dilution of *S. aureus* was seeded on tryptic soy agar (Hopebio) plates with 10 μg/ml nalidixic acid and incubated at 37 °C

for 6–8 h. Before use, all seeded plates were equilibrated at 25 °C for 1–2 h.

## *C. elegans* strains and culture

*C. elegans* strains were raised on NGM plates seeded with *E. coli* OP50 at 22 °C. For worm culture plates of defined pH, the pH of NGM and tryptic soy agar was adjusted with HCl prior to solidification. Transgenic strains were generated by microinjecting plasmid DNA, and at least two independent transgenic lines were tested to confirm the results. Integrated transgenic strains were created using gamma irradiation and were outcrossed at least three times before use. The strains used in this study are listed in Supplementary Data 1. All experiments were performed using young adult hermaphrodites unless specified otherwise.

## Mutant rescue and gene knockdown

Mutants were rescued by expressing wild-type cDNAs. Gene knockdown was achieved via RNAi by co-expressing two plasmids that encoded complementary sense and antisense mRNA fragments of the target gene. The promoters used for cell-specific rescue and RNAi included *Punc-17Δ1* (cholinergic VNC motor neurons)[39], *Pdel-1* (VA and VB motor neurons)[39], *Pges-1* (IECs)[54], and *Pdat-1* (CEP, ADE, and PDE neurons)[39]. Intestinal-specific RNAi of *aex-5* and *nhx-2* was done by feeding *C. elegans* strain VP303 with *E. coli* HT115 carrying L4440 plasmids containing target gene fragments[19]. All *C. elegans* cDNAs and target mRNA fragments were cloned from a Bristol N2 cDNA library. Mouse *Asic1a* and *Nhe1* cDNAs were amplified from a mouse kidney cDNA library. Primers (Tsingke Biotechnology Co., Ltd) are listed in Supplementary Table 1.

## Imaging

NHX-6::mStrawberry and ASIC-1::GFP translational fusion protein imaging was performed using an Olympus IXplore SpinSR confocal microscope with a 60×/1.42 NA oil immersion objective. All other fluorescence images were acquired using a Nikon Ts2R inverted microscope equipped with an Mshot MD60 CCD camera and the MShot Image Analysis System (version 1.1). Acquisition settings were kept constant throughout each experiment. For calcium and pHluorin imaging, young adult hermaphrodites were immobilized on coverslips using Vetbond Tissue Adhesive (3 M Company). Worms were bathed in a solution containing 140 mM NaCl, 5 mM KCl, 5 mM CaCl$_2$, 5 mM MgCl$_2$, 11 mM dextrose, and 5 mM HEPES (pH 7.2, 320 mOsm) and imaged at 1 frame per second for 2 min. For infection assays, young adult worms were cultivated on *S. aureus* for 4–6 h or *P. aeruginosa* for 8–12 h prior to imaging. For acid-evoked calcium imaging, because VNC motor neurons were inaccessible to bath-applied acid, a longitudinal incision was made along the glued region to expose neurons in the anterior part of the animal. Acidic solutions were prepared by adjusting the pH with HCl, and the osmolarity was adjusted to 320 mOsm. Acidic solutions were applied to the VNC via pressure ejection (2 psi for 5 s) at a 90° angle using an Eppendorf FemtoJet 4i microinjector.

## Whole-cell current recording

VA5 and VB6 motor neuron were identified based on their anatomical locations. For whole-cell current recordings[91], borosilicate glass pipettes (tip resistance ~20 MΩ) were used as electrodes. Classical whole-cell configuration was obtained by applying negative pressure to the recording pipette. Recordings were performed on a Nikon FN1 microscope with a MultiClamp 700B amplifier (Molecular Devices) using Clampex 10 (Molecular Devices). A series of voltage steps (−60 to +70 mV at 10-mV intervals and 1.2-s duration) was applied from a holding voltage of -60 mV. Data were sampled at a rate of 10 kHz after filtering at 2 kHz.

## Bacterial lawn avoidance assay

For avoidance assays[29], a 10 µl drop of overnight bacterial cultures was placed at the center of a 3.5-cm agar plate to make partial-lawn plates. Twenty L4 worms grown on *E. coli* OP50 were transferred to the center of each bacterial lawn and incubated at 25 °C. The number of worms on the bacterial lawns ($N_{on}$) was counted at designated time points after exposure. Percent avoidance = $(N_{total} − N_{on})/N_{total} × 100\%$. All experiments were independently repeated three times.

## Survival assay

For survival assays[90], 10 µl of overnight bacterial culture was spread to cover the entire surface of a 3.5-cm NGM plate to make full-lawn plates. To prevent progeny production and bagging, 5-fluoro-2′-deoxyuridine (FUdR, 50 µg/ml; Aladdin) was included. Synchronized L4 worms (100 per plate) were transferred onto the plates and cultivated at 25 °C. Worm viability was assessed every 12 h, and those unresponsive to nose and tail touch were scored as dead. Worms that died due to bursting vulva or crawled off the plate were censored from analysis. Live worms were transferred to fresh plates daily. Percent survival = $(N_{live}/N_{total}) × 100\%$. For oxotremorine treatment, 1 mM oxotremorine (Aladdin) was added to NGM plates. All experiments were independently repeated three times.

## Behavioral analysis

All behavioral assays were performed at 22 °C. Locomotion was recorded and analyzed using an automated *Track-A-Worm* system (version 2.0)[92]. A single young adult worm was transferred to the center of a 6-cm NGM plate seeded with a thin layer of designated bacteria. After 30 s for recovery, the worm was imaged for 1 min at 15 frames per second. Pharyngeal pumping, the rhythmic contraction and relaxation of the pharynx that drives bacterial ingestion, was manually quantified under a Nikon SMZ800N stereomicroscope. Pumping rate was calculated as the number of contractions of the pharyngeal terminal bulb in 60 s[93], serving as a quantitative measure of feeding activity. Counting was performed in triplicate and averaged for each worm. Defecation cycles were observed under the same stereomicroscope. The duration between contractions of the posterior body muscles was recorded 10 times and averaged for each worm[20].

## qRT-PCR analysis

Worms were collected in M9 buffer and lysed using TRIzol (TransGen Biotech, ET111-01). Total RNA was extracted by chloroform separation followed by isopropanol precipitation, washed with 75% ethanol, and resuspended in RNase-free water. cDNA was then synthesized using the HiScript II Q RT SuperMix kit (Vazyme, R233-01). qRT-PCR was conducted using SYBR qPCR Master Mix (Biosharp, BL697A) on a CFX Connect Real-Time PCR Detection System (Bio-Rad). mRNA levels were normalized using β-actin mRNA, and relative fold changes were calculated using the ΔΔCt method. All experiments were independently repeated three times. Primers (Tsingke Biotechnology Co., Ltd) are listed in Supplementary Table 2.

## Western blot

Day 1 adult worms were collected in M9 buffer and lysed by sonication in RIPA buffer (Beyotime, P0013C) supplemented with 1 mM PMSF (Biosharp, BL507A) and a protease/phosphatase inhibitor cocktail (NCM Biotech, P002). Protein levels of phosphorylated PMK-1, total PMK-1, and β-actin were analyzed by standard western blotting using rabbit anti-phospho-p38 MAPK monoclonal antibody (1:1000; ABclonal, AP0526), rabbit anti-p38 MAPK monoclonal antibody (1:1000; Abcam, ab170099), and mouse anti-β-actin monoclonal antibody (1:5000; ABclonal, AC004). Immunoblots were visualized using the ChemiScope 6000 system (Clinx), and band intensities were quantified using ImageJ (FIJI, version 1.50i). All experiments were independently repeated three times.

## Quantification of intestinal bacteria accumulation

To visualize and quantify intestinal accumulation of bacteria, full-lawn plates were prepared by spreading an overnight culture of OP50-GFP[94]. Synchronized L4 worms grown on *E. coli* OP50 were transferred to the plates and incubated at 25 °C for the indicated time. After washed with M9 buffer to remove any external bacteria, worms were imaged using the Nikon Ts2R inverted microscope.

Intestinal accumulation of *S. aureus* was quantified using colony-forming unit (CFU) assays[22]. L4 worms grown on *E. coli* OP50 were transferred to full-lawn *S. aureus* plates and incubated at 25 °C for the designated time. Fifty worms were picked into M9 containing 0.1 M sodium azide to be paralyzed. After washing three times with M9 containing 1 mg/ml ampicillin (Biosharp) and 1 mg/ml gentamicin (Biosharp), worms were incubated for 30 min in antibiotic M9 to remove any external bacteria. Worms were then lysed with a motorized pestle, serially diluted, and spread on TSA plates containing 10 μg/ml nalidixic acid. The plates were incubated overnight at 37 °C to determine CFUs.

## Data analysis

Calcium and pHluorin imaging data were analyzed using ImageJ, with individual cells assigned as separate regions of interest (ROIs). For VNC cholinergic motor neuron imaging, all neurons in the visual field were analyzed. For IEC imaging, the int2 cells were selected. For calcium imaging, fluorescence intensity (F) in each ROI was plotted as absolute intensity over time and then converted to a bleach-corrected ΔF/F0 values using a custom MATLAB module[39]. Spontaneous calcium transients were defined as events with $\Delta F/F0 \geq 0.05$ lasting ≥3 s in neurons and $\Delta F/F0 \geq 0.1$ lasting ≥10 s in IECs, and inter-peak intervals were calculated from recordings containing at least two detectable events. For pHluorin imaging, the co-expressed, pH-insensitive mStrawberry signal was used as an internal expression control to assess extracellular pH changes independently of potential differences in reporter expression.

Whole-cell current data were analyzed using Clampfit 10 (Molecular Devices). Current amplitudes were calculated from the mean current during the last 100 ms of each 1.2-s voltage pulse.

Statistical analyses and data graphing were performed using Origin 2019 (OriginLab), Prism 8 (GraphPad), or IBM SPSS Statistics 21 (IBM). One-way ANOVA with Tukey's post hoc test was used for multiple group comparisons, two-sided unpaired *t* test for two group comparisons, and log-rank (Kaplan–Meier) test for survival analyses. Behavioral avoidance and whole-cell current data were analyzed using two-way repeated-measures ANOVA. When a significant main effect or interaction was detected in the omnibus model, post hoc pairwise comparisons were performed using Sidak's multiple-comparisons correction. The figure legends provide detailed statistical information, including statistical methods, *n* numbers, and *p* values. Statistical significance was set at $p < 0.05$. No data were excluded from analyses, and all data were shown as mean ± SEM.

## Reporting summary

Further information on research design is available in the Nature Portfolio Reporting Summary linked to this article.

## Data availability

All data generated or analyzed during this study are included in this published article and its supplementary information files. Source data are provided with this paper.

## Code availability

The MATLAB module used to correct the photobleaching-induced drop of the fluorescence signal, along with user instructions, is freely available at https://health.uconn.edu/worm-lab/track-a-worm/.

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

## Acknowledgements

We would like to thank Dr. Bin Qi, Dr. Jianke Gong, and Dr. Taihong Wu for providing bacterial strains, and Dr. Jianke Gong, Dr. Donglei Zhang, and Dr. Wenxing Yang for plasmids. We also thank the Caenorhabditis Genetics Center (USA), which is funded by NIH Office of Research Infrastructure Programs (P40OD010440), for providing *E. coli* OP50 and *C. elegans* strains. This work was supported by the National Natural Science Foundation of China (32171003 and 32571188 to P.L.) and the Interdisciplinary Research Program of HUST (5003170102 to P.L.).

## Author contributions

Conceptualization, X.Z., C.C., and P.L.; experimentation and data analysis, Y. Lei, X.Z., Y. Liu, Y.W., and P.L.; writing, P.L.; administration, funding acquisition, resources, and supervision, P.L.

## Competing interests
The authors declare no competing interests.
