## [Transparent Peer Review file · Nature Communications]

Proton signaling links epithelial sensing to neural control of host defense in *C. elegans*

Corresponding Author: Dr Ping Liu

Version 0:

Reviewer comments:

Reviewer #1

(Remarks to the Author)

In this comprehensive and well-written manuscript, Lei et al provide convincing evidence for an important mechanism of pathogen detection by the intestinal epithelium in *C. elegans*. Prior work had shown that a cholinergic signal produced from the nervous system during infection triggered Wnt signaling in the intestinal epithelium, ultimately driving the expression of antimicrobial genes that promote survival of infection. However, the upstream signal that triggers the acetylcholine signal from neurons was unknown. The present work provides a plausible mechanism for at least part of the story.

The overall model proposed here is that intestinal distention and/or membrane disruption of the IECs produces a mechanical stimulus that drives GON-2-mediated Ca²⁺ influx. This influx activates NHX-6 via CMD-1 to export H⁺ through the basolateral side, acidifying the area surrounding the epithelium. Cholinergic motor neurons that surround the epithelium in close proximity use ASIC-1 to sense the pH in this space. Upon acidification, ASIC-1 triggers Ca²⁺ transients in these neurons, resulting in acetylcholine release, which through previously identified muscarinic receptors triggers both Wnt signaling and the p38 MAPK pathway in the intestinal epithelium.

Overall, the data appear compelling and the interpretation is mostly warranted. However, a few concerns should be addressed:

MAJOR

- The proposed model hinges on mechanosensation, but this is not proven or likely. Heat-killed *S. aureus* triggers a similar transcriptional response of live *S. aureus* (Irazoqui PNAS 2008), but the authors show that HK *S. aureus* does not trigger the GON-2 pathway. *Pseudomonas* does not induce gut distention until later in the infection (Aballay, Kim, others), which places that at odds with the kinetics observed here. Furthermore, the authors do not use latex beads, which distend the gut but are inert, to test this hypothesis. At least, the proposal for mechanosensation as THE trigger (or with membrane damage, see below) are merely speculative and should be couched with much more caution. Experiments to validate one or the other are preferred.
- No description of microfluidic device experiments for neuron GCAMP measurements. Without this description, it is not possible to interpret the signals from the different microbial triggers.
- No description of how IEC GCAMP was acquired. Similar to above.
- No description of pH control/rescue method. The lack of detail of how the pH was controlled and how the worms were exposed precludes any interpretation.
- They posit that membrane damage triggers GON-2, but if the membrane is damaged, cation chemical gradients with the extracellular space would be expected to dissipate. How do they reconcile? This is especially important for H⁺, which diffuse rapidly.
- Does the entire pseudocoelom acidify under conditions of infection?

MINOR

- Fig. 3b – the data for VB6 don't follow the color convention?
- Wrong statistics for Avoidance vs Time assays (Two-Way Repeated Measures ANOVA. If they wish to show differences at specific times (e.g., 24h), they should perform post-hoc pairwise comparisons (e.g., Bonferroni or Sidak corrected) only after finding a significant main effect or interaction in the omnibus model.)
- Discuss the benefit of having this IEC neuron IEC loop instead of cell-autonomous IEC signaling alone. Is it just for behavior?
- More discussion that the effects are partial – other missing parts/inputs?
- The transcriptional responses to *S. aureus* and *P. aeruginosa* are very different, yet both trigger this pathway. Discuss how specificity may be achieved.

Reviewer #2

(Remarks to the Author)

Host organisms must combat pathogens to ensure individual survival and fitness. Consequently, they have evolved a suite of defense strategies to systemically maintain physiological homeostasis and protect vital organ functions—a process fundamental to biological integrity and species stability. While the nervous system is known to play a critical role in coordinating host responses to infection, the underlying mechanisms remain poorly understood. In this study, the authors employ *C. elegans* as a model to elucidate a novel signaling pathway mediated by protons and acetylcholine that integrates behavioral avoidance with immune defense against pathogens such as *S. aureus* and *P. aeruginosa*. The authors demonstrate that ingested pathogens trigger proton release from intestine cells, which subsequently activates VA/VB motor neurons via the acid-sensing ion channel ASIC-1. This neuronal activation not only drives avoidance behavior but also initiates a feedback signal that potentiates intestinal immune-defense by activating the Wnt and p38/PMK-1 MAPK pathways. The study is supported by extensive data generated through multiple complementary techniques and the manuscript is well-organized, all of which make their findings robust and compelling.

Major:

- 1) What is the source of protons released from intestinal cells during infection? Specifically, does the release originate from lysosomal compartments, or is it generated through specific biochemical reactions triggered by ingested pathogens?
- 2) How does pathogen alter the calcium influx mediated by GON-2? Is the observed calcium influx increase a direct consequence of pathogen-induced mechanical stress? Is GON-2 mechanosensitive?
- 3) How do GON-2 and CMD-1 specifically regulate proton release from intestine cells?
- 4) What is the specific motor program executed by VA and AB neurons to drive pathogen avoidance, and is this circuit modulated by other neurons or neuroendocrine signals?
- 5) Does intestinal GON-2 mediate the pathogen-evoked shifts in basal calcium levels (measured by GCaMP6) in VA/VB sensory neurons?

Minor:

- 1) Is there any possibility that Calcium/Calmodulin might regulate PMK-1/P38 MAPK pathway cell-autonomously?
- 2) Is there any possibility that $\text{NHX-6} \text{---} \text{H}^+ \text{---} \text{ASIC-1}$ signaling pathway would regulate immune-defense temporally?

Reviewer #3

(Remarks to the Author)

The authors identify a role for a proton-mediated gut-to-neuron signaling pathway in the integration of behavioural and immune defenses against pathogen infection. They use an elegant combination of experimental approaches to provide substantial physiological and mechanistic insight into the components of this response and where and how they contribute. Importantly, they also show that the mammalian counterparts of the two key components of the pathway, an Na^+/H^+ exchanger and an acid sensing ion channel, can substitute for the *C. elegans* genes, underlining the high level of conservation and the relevance of a nematode model for studying gut-nervous system interactions. This is an important and topical area of research and of general interest. The research is of high quality, using innovative experimental approaches, well-founded on a thorough consideration of the current state of the field. It would thus be highly suitable for publication in this journal.

However, I have a number of serious concerns with missing controls, missing statistical comparisons and inappropriate analysis of the data, with the result that the conclusions are not always supported by the data. This must be addressed before it is suitable for publication. In most cases, this would be rectified by small changes to the way in which the current data is presented or analyzed, or by ensuring that their statements take account of the caveats.

I therefore strongly recommend publication, subject to revision to address these points.

Major comments

The authors use a pHluorin-mStrawberry fusion, quantifying fluorescence as a readout for pH in the extracellular environment of the motor neurons. The data appear to show pHluorin fluorescence, without, for example, use of the mStrawberry fluorescence as a control. The possibility thus remains that the observed change results from a change in expression levels of the pH indicator. Indeed, the authors use an *unc-17* promoter to drive its expression, but later show that *unc-17* is itself implicated in this pathway.

For the intestinal calcium imaging experiments (fig. 2d, e), the individual data points are discrete values, multiples of 0.5. This suggests that the data was analyzed by simply counting the number of peaks or full cycles within the 2-minute recording time. This presumably explains why the mean frequency implies a surprisingly short interval length (approx. 1.7 per minute for *wt* = 35 seconds!). The data would be more powerful, and the comparisons more meaningful, if the authors calculated the mean interval between peaks. The short recording interval, coupled with the small number of replicates,

means that the number of data points is small, particularly given that we know from previous studies that gon-2 loss of function results in increased variance of interval length (Xing et al. 2008, Kwan et al. 2008).

Figure 2e and related supplemental: We know from previous research that gon-2 loss of function results in an increase in cycle length, for both calcium and behavior (Xing et al. 2008). To convincingly say that gon-2 is required for the pathogen-induced change in calcium transient frequency and amplitude, animals grown on *E. coli* and pathogen must be compared directly. Given the role of gon-2 in contributing to the calcium events that play such a critical central role in both rhythmicity and overall function of the gut, care should be taken in determining whether the role identified here is specific to gon-2 or an overall contribution of intracellular calcium signaling.

Figure 2c, f, g: To convincingly say that these genes are required for the pathogen-induced change in pHluorin fluorescence, animals grown on *E. coli* and pathogen must be compared directly.

For the motoneuron silencing experiments using a histamine-gated chloride channel (supp. fig. 5), a control for the effects of histamine alone (i.e. in the absence of His-Cl1 expression) is required.

The authors show that, compared with *E. coli*, exposure to the pathogenic bacteria resulted in an increase in locomotion speed. They also show that on the pathogens *asic-1* and *nhx-6* mutants show a lower speed than wild type. To assert that the pathogen effect on speed is disrupted in these mutants, they need to show evidence that the change is disrupted, i.e. directly compare *E. coli*-grown with pathogen-grown.

Altering the growth medium pH is an appealing experiment and may support the conclusions about the role of ASIC-1. However, the observed effects could be due to any number of off-target effects unrelated to the extracellular environment of the motoneurons. Indeed, it is not clear how a change in pH in the medium would be translated into a corresponding change in an internal compartment, particularly given the important roles proton signaling/pH oscillations in multiple locations. The conclusions should not be overstated.

When using TRPV1 expression as a means to activate the motoneurons in an ASIC-1-independent manner (supp. Fig. 6h), the authors do not show any controls for the effects of capsaicin itself (i.e. whether in an *asic-1* loss of function background capsaicin has analogous effects, to exclude the possibilities that the observed effect is due to a *C. elegans* TRPV channel, which could be acting elsewhere).

Minor points

Line 111/Figure 1: The authors investigate the role of innexins expressed in the motor neurons, using data from Altun et al. 2009. More recent evidence (e.g. Bhattacharya et al. 2019; RNA seq data in the CeNGEN project (Taylor et al. 2021)) indicates a larger number of innexin genes are expressed in these cells. This should, at the very least, be acknowledged in the text. The authors also present no evidence that the RNAi actually knocks down the target genes or that it is expressed in the desired cells. Again, in the absence of controls, this should at least be acknowledged.

Line 145: "hours" does not seem to fit the sentence.

Figure 2e: *gtl-1* is incorrectly written as *glt-1*.

Figure 2d, e: n values are given for some genotypes but not for all.

Figure 4: misspelling of rescue.

Line 349: "pumping rates" requires more explanation for a non-specialist reader. Likewise, "increased colony-forming units" (or simply "CFUs" in the figure legend) – a better explanation of the experiment would be helpful.

Version 1:

Reviewer comments:

Reviewer #1

(Remarks to the Author)

The authors have appropriately addressed all my concerns.

Reviewer #2

(Remarks to the Author)

The authors have done a nice job in addressing my questions. It is ready for publication. Congratulations!

Reviewer #3

(Remarks to the Author)

All of my concerns have been addressed. I congratulate the authors on an excellent piece of work and have no further

comments.

Response to Reviewers' Comments

We thank the reviewers for their careful evaluation of our manuscript and for their constructive and insightful comments. We have addressed each point in detail, performed additional experiments, and revised the manuscript accordingly to strengthen the study. Below, we provide point-by-point responses. For clarity, reviewer comments are reproduced in *italics*, followed by our responses, with reference to the relevant text and figures in the revised manuscript. All changes have been incorporated into the revised manuscript and are highlighted in blue.

REVIEWER COMMENTS

Reviewer #1 (Remarks to the Author):

In this comprehensive and well-written manuscript, Lei et al provide convincing evidence for an important mechanism of pathogen detection by the intestinal epithelium in C. elegans. Prior work had shown that a cholinergic signal produced from the nervous system during infection triggered Wnt signaling in the intestinal epithelium, ultimately driving the expression of antimicrobial genes that promote survival of infection. However, the upstream signal that triggers the acetylcholine signal from neurons was unknown. The present work provides a plausible mechanism for at least part of the story.

The overall model proposed here is that intestinal distention and/or membrane disruption of the IECs produces a mechanical stimulus that drives GON-2-mediated Ca²⁺ influx. This influx activates NHX-6 via CMD-1 to export H⁺ through the basolateral side, acidifying the area surrounding the epithelium. Cholinergic motor neurons that surround the epithelium in close proximity use ASIC-1 to sense the pH in this space. Upon acidification, ASIC-1 triggers Ca²⁺ transients in these neurons, resulting in acetylcholine release, which through previously identified muscarinic receptors triggers both Wnt signaling and the p38 MAPK pathway in the intestinal epithelium.

Overall, the data appear compelling and the interpretation is mostly warranted. However, a few concerns should be addressed:

Response: We thank the reviewer for the overall positive assessment of our work.

MAJOR

- The proposed model hinges on mechanosensation, but this is not proven or likely. Heat-killed Saureus triggers a similar transcriptional response of live S. aureus (Irazoqui PNAS 2008), but the authors show that HK S. aureus does not trigger the*

GON-2 pathway. Pseudomonas does not induce gut distention until later in the infection (Aballay, Kim, others), which places that at odds with the kinetics observed here. Furthermore, the authors do not use latex beads, which distend the gut but are inert, to test this hypothesis. At least, the proposal for mechanosensation as THE trigger (or with membrane damage, see below) are merely speculative and should be couched with much more caution. Experiments to validate one or the other are preferred.

Response: We thank the reviewer for this important critique and agree that mechanosensation should not be presented as a single or definitive trigger of intestinal epithelial activation. Rather, our data support that pathogen infections enhance GON-2-dependent and mechanically sensitive Ca²⁺ signaling in intestinal epithelial cells (IECs).

Our conclusions regarding mechanical sensitivity are based on direct experimental evidence. In wild-type animals fed *E. coli*, suppressing body movement, either by physical immobilization with glue or by pharmacological paralysis with levamisole, significantly reduced spontaneous IEC Ca²⁺ transients, phenocopying *gon-2* mutants. Immobilization did not further suppress Ca²⁺ activity in *gon-2* mutants, and IEC-specific expression of *gon-2* rescued the defects (Supplementary Fig. 3d), suggesting that GON-2 is required for mechanically sensitive Ca²⁺ signaling associated with body bending.

During pathogen infections, infection-enhanced Ca²⁺ influx in IECs was similarly suppressed by immobilization in wild type, and immobilization of infected *gon-2* mutants produced no additional effect (Fig. 2e and Supplementary Fig. 3c). Together, these epistasis experiments suggest that pathogens potentiate an existing GON-2-dependent mechanosensitive signaling pathway.

The reviewer notes that heat-killed *S. aureus* induces transcriptional responses similar to live *S. aureus* (Irazoqui, J. E. et al., Plos Pathog, 2010), whereas we observe that heat-killed *S. aureus* fails to induce GON-2-dependent Ca²⁺ signaling. The transcriptional responses described by Irazoqui and colleagues are interpreted as PAMP-driven host responses. Our findings therefore suggest that live pathogens engage an additional layer of epithelial signaling, namely enhanced Ca²⁺ influx, likely reflecting infection-associated epithelial stress or damage rather than PAMP recognition alone.

We agree that bulk intestinal distention alone is unlikely to explain our observations. Consistently, genetically induced intestinal bloating in *E. coli*-fed animals via *aex-5* or *nhx-2* RNAi did not increase IEC Ca²⁺ transients

(Supplementary Fig. 3f). Although we did not use inert latex beads, these genetic manipulations provide an independent means of uncoupling intestinal distention from infection-induced Ca^{2+} signaling.

Together, our data support a model in which pathogen infections enhance GON-2-dependent mechanosensitive Ca^{2+} influx in IECs, likely reflecting infection-associated epithelial stress or membrane perturbation. This interpretation is consistent with current models of mechanosensitive channel gating, in which channel activity can be modulated by membrane tension, curvature, or lipid perturbations (Kefauver, J. M., Ward, A. B. & Patapoutian, A., *Nature*, 2020). However, we do not directly measure membrane tension or other biophysical parameters in this study. Accordingly, we have revised Fig. 7, its legend, and the Discussion (the 9th paragraph) to clarify that infection-associated epithelial stress is proposed to engage GON-2 indirectly, rather than invoking a defined or exclusive mechanical trigger.

• *No description of microfluidic device experiments for neuron GCaMP measurements. Without this description, it is not possible to interpret the signals from the different microbial triggers.*

Response: We thank the reviewer for raising this point. We did not use microfluidic devices for neuronal GCaMP imaging in this study. Instead, calcium and pHluorin imaging were performed on immobilized animals under defined bath conditions.

To clarify this, we have expanded the Methods (“Imaging”) to explicitly describe the imaging setup, immobilization procedure, bath composition, acquisition parameters, and infection conditions used for neuronal and intestinal imaging, as follows: “All other fluorescence images were acquired using a Nikon Ts2R inverted microscope equipped with an Mshot MD60 CCD camera and the MShot Image Analysis System (version 1.1). Acquisition settings were kept constant throughout each experiment. For calcium and pHluorin imaging, young adult hermaphrodites were immobilized on coverslips using Vetbond Tissue Adhesive (3M Company). Worms were bathed in a solution containing 140 mM NaCl, 5 mM KCl, 5 mM CaCl_2 , 5 mM MgCl_2 , 11 mM dextrose, and 5 mM HEPES (pH 7.2, 320 mOsm) and imaged at 1 frame per second for 2 min. For infection assays, young adult worms were cultivated on *S. aureus* for 4-6 hours or *P. aeruginosa* for 8-12 hours prior to imaging.”

In addition, we have clarified ROI selection and data analysis in the Methods (“Data analysis”) as follows: “Calcium and pHluorin imaging data were analyzed using ImageJ, with individual cells assigned as separate regions of interest (ROIs). For VNC cholinergic motor neuron imaging, all neurons in the visual field were analyzed. For IEC imaging, the int2 cells were selected. For calcium imaging,

fluorescence intensity (F) in each ROI were plotted as absolute intensity over time and then converted to a bleach-corrected $\Delta F/F_0$ values using a custom MATLAB module. Spontaneous calcium transients were defined as events with $\Delta F/F_0 \geq 0.05$ lasting ≥ 3 s in neurons and $\Delta F/F_0 \geq 0.1$ lasting ≥ 10 s in IECs, and inter-peak intervals were calculated from recordings containing at least two detectable events.”

• *No description of how IEC GCaMP was acquired. Similar to above.*

Response: We thank the reviewer for this comment. As noted above, we have expanded the Methods (“Imaging” and “Data analysis”) to provide a detailed description of the calcium imaging protocol used for both neuronal and IEC GCaMP imaging.

• *No description of pH control/rescue method. The lack of detail of how the pH was controlled and how the worms were exposed precludes any interpretation.*

Response: We thank the reviewer for this comment. To test motor neuron acid sensitivity, acidic solutions were applied directly to ventral nerve cord (VNC) motor neurons in dissected worms.

These procedures are now described in the revised Methods as follows “For calcium and pHluorin imaging, young adult hermaphrodites were immobilized on coverslips using Vetbond Tissue Adhesive (3M Company). Worms were bathed in a solution containing 140 mM NaCl, 5 mM KCl, 5 mM CaCl_2 , 5 mM MgCl_2 , 11 mM dextrose, and 5 mM HEPES (pH 7.2, 320 mOsm) and imaged at 1 frame per second for 2 min.” and “For acid-evoked calcium imaging, because VNC motor neurons were inaccessible to bath-applied acid, a longitudinal incision was made along the glued region to expose neurons in the anterior part of the animal. Acidic solutions were prepared by adjusting the pH with HCl, and the osmolarity was adjusted to 320 mOsm. Acidic solutions were applied to the VNC via pressure ejection (2 psi for 5 s) at a 90° angle using an Eppendorf FemtoJet 4i microinjector.”

To examine the effects of exogenous H^+ on pathogen avoidance and survival, nematode growth medium (NGM) or tryptic soy agar plates were acidified by HCl adjustment prior to solidification, generating plates with defined pH values (pH 4-7). These procedures are now described in the revised Methods as follows “For worm culture plates of defined pH, the pH of nematode growth medium (NGM) and tryptic soy agar was adjusted with HCl prior to solidification.”

• *They posit that membrane damage triggers GON-2, but if the membrane is damaged, cation chemical gradients with the extracellular space would be expected to dissipate. How do they reconcile? This is especially important for H^+ , which diffuse*

rapidly.

Response: We thank the reviewer for this thoughtful critique and agree that widespread or sustained membrane rupture would be expected to dissipate ionic gradients, particularly for highly diffusible ions such as H⁺. In our model, however, by “membrane perturbation” or “epithelial stress”, we refer to localized or sublethal alterations in membrane tension or integrity that can arise during pathogen infections, rather than gross membrane rupture.

Consistently, our data show that pathogen infections enhance GON-2-dependent Ca²⁺ influx in IECs without causing indiscriminate Ca²⁺ entry or baseline depolarization under our experimental conditions, arguing against nonspecific membrane leakage. We therefore propose that GON-2 activation occurs within a physiological regime in which localized membrane perturbation or epithelial stress enhances mechanosensitive Ca²⁺ influx while overall ionic homeostasis is preserved, likely supported by the rapid repair capacity of the intestinal epithelium. Such signaling is likely to be spatially and temporally restricted, occurring prior to severe epithelial compromise.

We have revised the Discussion as follows: “This infection-associated enhancement likely reflects localized epithelial stress or membrane perturbation arising during infections, operating within a spatially and temporally restricted physiological regime that precedes severe epithelial compromise.”

• *Does the entire pseudocoelom acidify under conditions of infection?*

Response: We thank the reviewer for this important question. In our initial experiments, we recorded pHluorin signals from VNC motor neurons in the mid-anterior region of the animal, due to the limited field of view required for high-resolution imaging. To address whether infections induce widespread pseudocoelomic acidification, we extended our analysis to 6 additional VNC motor neurons located more distally, including anterior (DB3, DA2, VA3, VB4) and posterior (VA10, VB11) neurons.

Although these experiments do not directly determine whether the entire pseudocoelom becomes acidified, they reveal extracellular acidification at widespread VNC locations during pathogen infections (new Supplementary Fig. 3a). We have incorporated these new findings into the revised Results as follows: “Although these experiments do not directly determine whether pathogen infections cause global pseudocoelomic acidification, they demonstrate that the extracellular space surrounding motor neurons at widespread VNC sites becomes acidified during pathogen infections.”

MINOR

- *Fig. 3b – the data for VB6 don't follow the color convention?*

Response: Corrected, thanks.

- *Wrong statistics for Avoidance vs Time assays (Two-Way Repeated Measures ANOVA. If they wish to show differences at specific times (e.g., 24h), they should perform post-hoc pairwise comparisons (e.g., Bonferroni or Sidak corrected) only after finding a significant main effect or interaction in the omnibus model.)*

Response: We thank the reviewer for this clarification. We have reanalyzed the avoidance data using two-way repeated-measures ANOVA. When a significant main effect or interaction was detected in the omnibus model, post hoc pairwise comparisons were performed using Sidak's multiple-comparisons correction. The updated statistical analyses are now reported in revised Fig. 4c-g, Fig. 6e, Supplementary Fig. 7b-h, and Supplementary Fig. 11f. The corresponding figure legends and Methods have been updated accordingly.

- *Discuss the benefit of having this IEC □ neuron □ IEC loop instead of cell-autonomous IEC signaling alone. Is it just for behavior?*

Response: We thank the reviewer for this important suggestion. We propose that the IEC-neuron-IEC loop provides advantages beyond cell-autonomous IEC signaling by enabling rapid coordination of behavioral and immune responses. By engaging the nervous system, intestinal epithelial sensing can simultaneously promote pathogen avoidance behavior, thereby limiting pathogen ingestion, while enhancing intestinal defense. In addition, this circuit architecture likely enables signal amplification, flexibility, and integration with internal state, features that would be difficult to achieve through IEC-intrinsic signaling alone. We have incorporated this discussion into the revised Discussion as follows: "By routing epithelial sensing signals through the nervous system, the IEC-neuron-IEC loop enables rapid coupling of behavioral avoidance with intestinal immunity, while affording amplification, flexibility, and integration with internal state that cannot be achieved by cell-autonomous IEC signaling alone."

- *More discussion that the effects are partial – other missing parts/inputs?*

Response: We thank the reviewer for this insightful comment and agree that the NHX-6-H⁺-ASIC-1 pathway represents a partial component of host defense and does not act in isolation. Our data indicate that this pathway enhances, rather than solely triggers, pathogen avoidance and intestinal innate immunity. Additional inputs are likely to contribute, including cholinergic signaling from additional neurons, other

neuronal pathways, established pathogen-sensing and immune signaling mechanisms, and parallel epithelial defense pathways. We have clarified in the revised Discussion as follows: “These findings define a form of damage-associated immune surveillance that links epithelial sensing to neural control of host defense, functioning alongside established pathogen-sensing pathways, other neuronal inputs, and parallel epithelial immune mechanisms.”

• *The transcriptional responses to S.aureus and P. aeruginosa are very different, yet both trigger this pathway. Discuss how specificity may be achieved.*

Response: We thank the reviewer for raising this insightful comment. Although *S. aureus* and *P. aeruginosa* elicit substantially different transcriptional immune responses, our data suggest that proton-mediated gut-to-neuron signaling reflects a shared physiological readout of intestinal epithelial stress rather than pathogen-specific recognition. We propose that this pathway functions as a common, rapid alert signal that coordinates behavioral avoidance and immune responses, while pathogen specificity is achieved through parallel signaling pathways that engage distinct pattern- or damage-associated cues and drive pathogen-biased transcriptional responses. We have incorporated these into the revised Discussion as follows: “Although *S. aureus* and *P. aeruginosa* elicit divergent transcriptional responses, proton-mediated gut-to-neuron signaling likely represents a shared physiological readout of intestinal epithelial stress, while pathogen specificity is conferred by parallel signaling pathways that drive distinct transcriptional and immune programs.”

Reviewer #2 (Remarks to the Author):

Host organisms must combat pathogens to ensure individual survival and fitness. Consequently, they have evolved a suite of defense strategies to systemically maintain physiological homeostasis and protect vital organ functions—a process fundamental to biological integrity and species stability. While the nervous system is known to play a critical role in coordinating host responses to infection, the underlying mechanisms remain poorly understood. In this study, the authors employ C. elegans as a model to elucidate a novel signaling pathway mediated by protons and acetylcholine that integrates behavioral avoidance with immune defense against pathogens such as S. aureus and P. aeruginosa. The authors demonstrate that ingested pathogens trigger proton release from intestine cells, which subsequently activates VA/VB motor neurons via the acid-sensing ion channel ASIC-1. This neuronal activation not only drives avoidance behavior but also initiates a feedback

signal that potentiates intestinal immune-defense by activating the Wnt and p38/PMK-1 MAPK pathways. The study is supported by extensive data generated through multiple complementary techniques and the manuscript is well-organized, all of which make their findings robust and compelling.

Response: We thank the reviewer for the overall positive assessment of our work.

Major:

1) What is the source of protons released from intestinal cells during infection? Specifically, does the release originate from lysosomal compartments, or is it generated through specific biochemical reactions triggered by ingested pathogens?

Response: We thank the reviewer for this insightful question. Our data demonstrate that pathogen infections induce NHX-6-dependent proton release from intestinal epithelial cells (IECs). However, the immediate intracellular source of these protons cannot be conclusively resolved by our experiments.

C. elegans IECs maintain dynamic proton pools associated with rhythmic intestinal pH oscillations, driven in part by proton movement across the apical membrane from the intestinal lumen (Pfeiffer et al., 2008). Pathogen exposure may therefore promote basolateral release of pre-existing proton gradients. As the reviewer noted, it is also possible that intracellular acidic compartments contribute. Notably, lysosome-related organelles in IECs are disrupted by pathogen-derived toxins such as pyocyanin from *P. aeruginosa* (Tse-Kang et al., 2024), raising the possibility that infection-associated epithelial stress could influence intracellular proton handling.

However, lysosomal proton efflux mechanisms remain poorly defined in *C. elegans*, and key lysosomal proton channels such as TMEM175 lack clear nematode orthologs (Hu et al., 2022). Thus, our findings define the signaling and transport machinery required for pathogen-induced proton release, while leaving open the precise subcellular origin of the protons involved.

We have clarified this point in the revised Discussion as follows “While our data demonstrate that pathogen infections trigger NHX-6-dependent proton release from IECs, the immediate intracellular source of these protons cannot be conclusively resolved by our experiments. IECs in *C. elegans* maintain dynamic proton pools associated with rhythmic intestinal pH oscillations, which are driven in part by proton movement across the apical membrane from the intestinal lumen. Pathogen exposure may therefore promote basolateral release of pre-existing proton gradients. In addition, intracellular acidic compartments could contribute to this response. Lysosome-related organelles in IECs are disrupted by pathogen-derived toxins such

as pyocyanin from *P. aeruginosa*, raising the possibility that infection-associated stress alters intracellular proton handling. However, lysosomal proton efflux mechanisms remain poorly defined in *C. elegans*, and key lysosomal proton channels such as TMEM175 lack clear nematode orthologs. Thus, our findings define the signaling and transport machinery required for pathogen-evoked proton release, while leaving open the precise subcellular origin of the protons involved.”

2) How does pathogen alter the calcium influx mediated by GON-2? Is the observed calcium influx increase a direct consequence of pathogen-induced mechanical stress? Is GON-2 mechanosensitive?

Response: We thank the reviewer for this important question. Our data show that pathogen infections enhance IEC Ca²⁺ influx in a GON-2-dependent manner. While this enhancement is consistent with increased mechanosensitive signaling, we do not claim it as a direct or single mechanical stimulus. Rather, we propose that infection-associated epithelial stress, such as localized changes in membrane tension or integrity, enhances mechanosensitive Ca²⁺ influx. This interpretation is consistent with current models of mechanosensitive channel regulation, in which channel activity can be modulated by membrane tension, curvature, or lipid perturbations (Suchyna, T. M., *Prog Biophys Mol Biol*, 2017), However, we do not directly measure membrane tension or other biophysical parameters in this study. We have revised Fig. 7, its legend, and the Discussion (the 9th paragraph) to reflect this limitation and to avoid overinterpretation.

Moreover, although our genetic and epistasis analyses demonstrate that GON-2 mediates mechanosensitive Ca²⁺ influx in IECs during body bending, they do not establish whether GON-2 itself is the pore-forming mechanosensitive channel. GON-2 may instead function as an essential component or regulator of a mechanosensitive channel complex, or influence channel expression, assembly, or localization. We have explicitly acknowledged this point in the revised Discussion as follows: “However, whether GON-2 itself constitutes the mechanosensitive pore-forming channel remains an open question. GON-2 may instead function as an essential component or regulator of a mechanosensitive channel complex, or influence channel expression, assembly, or localization.”

3) How do GON-2 and CMD-1 specifically regulate proton release from intestine cells?

Response: We thank the reviewer for this important question. Our data indicate that GON-2-dependent Ca²⁺ influx and the downstream Ca²⁺ sensor CMD-1 are required to engage NHX-6-dependent proton release from IECs during pathogen infections.

However, we do not propose that GON-2 and CMD-1 function exclusively in proton signaling. Rather, they may act as upstream epithelial stress sensors that couple Ca^{2+} signaling to multiple downstream responses relevant to host defense, with proton release representing one physiologically important output that enables gut-to-neuron communication.

We have highlighted this in the revised Discussion as follows: “In parallel, Ca^{2+} /calmodulin signaling may also directly engage PMK-1/p38 MAPK or other immune signaling pathways cell-autonomously in IECs, as has been demonstrated for p38 MAPK regulation in AWC olfactory sensory neurons, providing an additional layer of regulation that complements gut-to-neuron communication.” and “These findings define a form of damage-associated immune surveillance that links epithelial sensing to neural control of host defense, functioning alongside established pathogen-sensing pathways, other neuronal inputs, and parallel epithelial immune mechanisms.”

4) What is the specific motor program executed by VA and AB neurons to drive pathogen avoidance, and is this circuit modulated by other neurons or neuroendocrine signals?

Response: We thank the reviewer for this important question. A-type (including VA and DA) and B-type (including VB and DB) ventral nerve cord motor neurons are cholinergic neurons that drive backward (reversal) and forward locomotion, respectively, with VA/VB innervating ventral muscles and DA/DB innervating dorsal muscles. These motor neurons receive inputs from command interneurons that integrate sensory and internal-state information to bias locomotor output.

Pathogen avoidance involves changes in reversal frequency, turning, and locomotion speed, reflecting shifts in motor patterns. Our data indicate that pathogen-induced activation of VA and VB promotes pathogen avoidance by increasing average, forward, and backward speeds, and that these effects require NHX-6 in IECs and ASIC-1 in VA/VB (Fig. 4a,b; Supplementary Fig. 7a).

VA and VB neurons are embedded within the locomotor circuit and are likely modulated by upstream interneurons and neuromodulatory or neuroendocrine signals during pathogen infections. Dissecting these additional circuit-level contributions will be an important direction for future work. We have clarified in the revised Discussion as follows: “Within this framework, proton-mediated activation of motor neurons enhances locomotor dynamics associated with pathogen avoidance, while remaining embedded within a broader motor circuit subject to modulation by upstream interneurons and neuromodulatory signals.”

5) Does intestinal GON-2 mediate the pathogen-evoked shifts in basal calcium levels (measured by GCaMP6) in VA/VB sensory neurons?

Response: We thank the reviewer for this thoughtful question. To directly address this, we performed additional GCaMP6 imaging. We found that *gon-2(q362)* mutants exhibited significantly reduced basal GCaMP6 fluorescence and spontaneous Ca²⁺ transient amplitude in VA/VB during pathogen infections, and these defects were rescued by IEC-specific *gon-2* expression. Moreover, the *gon-2(q362)* mutation did not further reduce VA/VB Ca²⁺ activity in *nhx-6(ok609)* mutants (new Supplementary Fig. 5c-f), consistent with GON-2 and NHX-6 acting in the same pathway. Together, these results suggest that intestinal GON-2 is required to couple pathogen infections to elevated VA/VB activity. We have incorporated these new findings into the revised Results.

Minor:

1) Is there any possibility that Calcium/Calmodulin might regulate PMK-1/P38 MAPK pathway cell-autonomously?

Response: We thank the reviewer for this insightful question. Ca²⁺/calmodulin-dependent regulation of p38 MAPK signaling has been described in other contexts. For example, in AWC olfactory neurons, Ca²⁺ influx through the UNC-2/UNC-36 calcium channel activates the Ca²⁺/calmodulin-dependent kinase UNC-43, which in turn engages p38 MAPK signaling during asymmetric cell fate specification (Sagasti et al., Cell, 2001).

These findings raise the possibility that Ca²⁺/calmodulin could also regulate PMK-1/p38 MAPK or other immune signaling pathways cell-autonomously in IECs. Although our data do not directly test this possibility, we have acknowledged it in the revised Discussion as follows: "In parallel, Ca²⁺/calmodulin signaling may also directly engage PMK-1/p38 MAPK or other immune signaling pathways cell-autonomously in IECs, as has been demonstrated for p38 MAPK regulation in AWC olfactory sensory neurons, providing an additional layer of regulation that complements gut-to-neuron communication."

2) Is there any possibility that NHX-6 — H⁺ — ASIC-1 signaling pathway would regulate immune-defense temporally?

Response: We thank the reviewer for this thoughtful question. Yes, our data support the idea that the NHX-6-H⁺-ASIC-1 pathway could function in a temporally restricted manner. We propose that this pathway is engaged early during pathogen infections, when IECs experience localized or sublethal epithelial stress. In contrast, widespread

or sustained membrane rupture would be expected to dissipate ionic gradients, particularly for highly diffusible ions such as H⁺, thereby limiting this mode of signaling

We have clarified in the revised Discussion as follows: “This infection-associated enhancement likely reflects localized epithelial stress or membrane perturbation arising during infections, operating within a spatially and temporally restricted physiological regime that precedes severe epithelial compromise.”

Reviewer #3 (Remarks to the Author):

The authors identify a role for a proton-mediated gut-to-neuron signaling pathway in the integration of behavioural and immune defenses against pathogen infection. They use an elegant combination of experimental approaches to provide substantial physiological and mechanistic insight into the components of this response and where and how they contribute. Importantly, they also show that the mammalian counterparts of the two key components of the pathway, an Na⁺/H⁺ exchanger and an acid sensing ion channel, can substitute for the C. elegans genes, underlining the high level of conservation and the relevance of a nematode model for studying gut-nervous system interactions. This is an important and topical area of research and of general interest. The research is of high quality, using innovative experimental approaches, well-founded on a thorough consideration of the current state of the field. It would thus be highly suitable for publication in this journal.

However, I have a number of serious concerns with missing controls, missing statistical comparisons and inappropriate analysis of the data, with the result that the conclusions are not always supported by the data. This must be addressed before it is suitable for publication. In most cases, this would be rectified by small changes to the way in which the current data is presented or analyzed, or by ensuring that their statements take account of the caveats.

I therefore strongly recommend publication, subject to revision to address these points.

Response: We thank the reviewer for the overall positive assessment of our work.

Major comments

The authors use a pHluorin-mStrawberry fusion, quantifying fluorescence as a readout for pH in the extracellular environment of the motor neurons. The data appear to show pHluorin fluorescence, without, for example, use of the mStrawberry

fluorescence as a control. The possibility thus remains that the observed change results from a change in expression levels of the pH indicator. Indeed, the authors use an unc-17 promoter to drive its expression, but later show that unc-17 is itself implicated in this pathway.

Response: We thank the reviewer for raising this important point. To address this concern, we performed additional control experiments by recording mStrawberry fluorescence in the same *Punc-17*-driven pHluorin-mStrawberry transgenic worms under the relevant infection and genetic conditions.

Across these conditions, mStrawberry fluorescence remained unchanged, indicating stable reporter expression. We have presented these control data in new Supplementary Fig. 4a-d and Supplementary Fig. 11c and described in the revised Results as follows:

“Across these conditions, the co-expressed, pH-insensitive mStrawberry signal remained unchanged during pathogen exposure (Supplementary Fig. 4a), indicating stable reporter expression.”

“mStrawberry fluorescence was unaffected by pathogen exposure across these genotypes (Supplementary Fig. 4b).”

“mStrawberry fluorescence was unaffected by pathogen exposure across these genotypes (Supplementary Fig. 4c,d).”

“mStrawberry fluorescence was unaffected across these rescue conditions (Supplementary Fig. 11c).”

We have also clarified this approach in the revised Methods (“Imaging”) as follows: “For pHluorin imaging, the co-expressed, pH-insensitive mStrawberry signal was used as an internal expression control to assess extracellular pH changes independently of potential differences in reporter expression.”

For the intestinal calcium imaging experiments (fig. 2d, e), the individual data points are discrete values, multiples of 0.5. This suggests that the data was analyzed by simply counting the number of peaks or full cycles within the 2-minute recording time. This presumably explains why the mean frequency implies a surprisingly short interval length (approx. 1.7 per minute for wt = 35 seconds!). The data would be more powerful, and the comparisons more meaningful, if the authors calculated the mean interval between peaks. The short recording interval, coupled with the small number of replicates, means that the number of data points is small, particularly given that we know from previous studies that gon-2 loss of function results in increased variance of interval length (Xing et al. 2008, Kwan et al. 2008).

Response: We thank the reviewer for this insightful and constructive comment. The reviewer is correct that, in our original analysis, Ca²⁺ activity in intestinal epithelial cells (IECs) was quantified over a 2-min recording at 1 frame per second, resulting in discrete values that reflect the number of Ca²⁺ transients detected within the imaging window. In response to this suggestion, we have reanalyzed the data by calculating the mean inter-peak interval for each animal, cited previous studies, and have revised the figures and text accordingly (revised Fig. 2d,e and Supplementary Fig. 3c-f). This analysis confirms our conclusions regarding GON-2-dependent enhancement of IEC mechanosensitive activation during pathogen infections.

Figure 2e and related supplemental: We know from previous research that gon-2 loss of function results in an increase in cycle length, for both calcium and behavior (Xing et al. 2008). To convincingly say that gon-2 is required for the pathogen-induced change in calcium transient frequency and amplitude, animals grown on E. coli and pathogen must be compared directly. Given the role of gon-2 in contributing to the calcium events that play such a critical central role in both rhythmicity and overall function of the gut, care should be taken in determining whether the role identified here is specific to gon-2 or an overall contribution of intracellular calcium signaling.

Response: We thank the reviewer for this important comment. Consistent with previous findings (Xing et al., 2008), *gon-2(q362)* mutants exhibited reduced intestinal Ca²⁺ activity, which was rescued by IEC-specific expression of *gon-2* (Supplementary Fig. 3d). As suggested, we directly compared *gon-2(q362)* mutants grown on *E. coli* or exposed to either *S. aureus* or *P. aeruginosa*. In contrast to wild-type animals (Fig. 2d), pathogen exposure did not significantly alter Ca²⁺ transient frequency, amplitude, or inter-peak interval in *gon-2(q362)* mutants (new Supplementary Fig. 3e). These results suggest that pathogen-induced enhancement of IEC Ca²⁺ signaling requires GON-2, while not excluding potential contributions from other sources of intracellular Ca²⁺ signaling.

We have incorporated these analyses and cited previous studies in the revised Results (“Pathogen-induced IEC proton secretion requires TRP channel GON-2-mediated Ca²⁺ influx and CMD-1/calmodulin”). We have also clarified this point in the revised Discussion as follows “we find that these phylogenetically distinct pathogens enhance IEC mechanosensitive Ca²⁺ influx via the TRP channel GON-2, while not excluding contributions from other sources of intracellular Ca²⁺ signaling”.

Figure 2c, f, g: To convincingly say that these genes are required for the pathogen-induced change in pHluorin fluorescence, animals grown on E. coli and pathogen must be compared directly.

Response: We thank the reviewer for this insightful suggestion. To address this, we performed additional pHluorin imaging using animals raised on *E. coli*, including *nhx-6*, *gon-2*, *nhx-6;gon-2*, *cmd-1(RNAi)*, and *nhx-6;cmd-1(RNAi)*. These new data enable direct within-genotype comparisons and show that loss of *nhx-6*, *gon-2*, or *cmd-1* abolishes the pathogen-induced reduction in pHluorin fluorescence (new Supplementary Fig. 4e). We have incorporated these new findings into the revised Results.

For the motoneuron silencing experiments using a histamine-gated chloride channel (supp. fig. 5), a control for the effects of histamine alone (i.e. in the absence of HisCl1 expression) is required.

Response: We thank the reviewer for this insightful suggestion. To address this concern, we performed additional control experiments comparing wild-type animals with and without histamine, as well as animals expressing HisCl1 in body-wall muscles in the absence of histamine. These new data show that neither histamine treatment alone nor HisCl1 expression alters motor neuron activity (new Supplementary Fig. 6c,d). We have incorporated these new findings into the revised Results.

*The authors show that, compared with E. coli, exposure to the pathogenic bacteria resulted in an increase in locomotion speed. They also show that on the pathogens *asic-1* and *nhx-6* mutants show a lower speed than wild type. To assert that the pathogen effect on speed is disrupted in these mutants, they need to show evidence that the change is disrupted, i.e. directly compare E. coli-grown with pathogen-grown.*

Response: We thank the reviewer for this important suggestion. To address this, we performed additional recordings using animals of each genotype raised on *E. coli*. These new data enable direct within-genotype comparisons and show that the pathogen-induced increase in locomotion speed observed in wild type is absent in *asic-1*, *nhx-6*, and *asic-1;nhx-6* mutants (revised Fig. 4b and Supplementary Fig. 7a). We have incorporated these new findings into the revised Results.

Altering the growth medium pH is an appealing experiment and may support the conclusions about the role of ASIC-1. However, the observed effects could be due to any number of off-target effects unrelated to the extracellular environment of the motoneurons. Indeed, it is not clear how a change in pH in the medium would be translated into a corresponding change in an internal compartment, particularly given the important roles proton signaling/pH oscillations in multiple locations. The conclusions should not be overstated.

Response: We thank the reviewer for raising this important point and agree that

altering the pH of the growth medium represents an indirect manipulation and may influence internal proton dynamics through multiple physiological processes. Accordingly, we interpret these experiments as supportive, correlative evidence consistent with a role for ASIC-1 in proton sensing, rather than as a precise manipulation of the motor neuron extracellular environment. We have clarified this interpretation in the revised Results as follows: “Although altering the growth medium pH is an indirect manipulation, these findings are consistent with ASIC-1-dependent proton sensing.”

When using TRPV1 expression as a means to activate the motoneurons in an ASIC-1-independent manner (supp. Fig. 6h), the authors do not show any controls for the effects of capsaicin itself (i.e. whether in an asic-1 loss of function background capsaicin has analogous effects, to exclude the possibilities that the observed effect is due to a C. elegans TRPV channel, which could be acting elsewhere.

Response: We thank the reviewer for this important suggestion. Our existing data show that capsaicin treatment alone does not affect avoidance behavior in wild-type animals. To further exclude nonspecific effects of capsaicin, activation of endogenous *C. elegans* TRPV channels, or TRPV1 expression, we performed additional control experiments.

We find that capsaicin treatment alone also has no effect on avoidance in *asic-1* mutants, and that expression of TRPV1 in VA/VB neurons in the absence of capsaicin does not alter avoidance behavior (revised Supplementary Fig. 7h). Together, these controls demonstrate that the observed effects are specific to capsaicin-dependent activation of heterologously expressed TRPV1 in motor neurons, rather than off-target effects of capsaicin, endogenous channels, or TRPV1 expression. We have incorporated these new findings into the revised Results.

Minor points

Line 111/Figure 1: The authors investigate the role of innexins expressed in the motor neurons, using data from Altun et al. 2009. More recent evidence (e.g. Bhattacharya et al. 2019; RNA seq data in the CeNGEN project (Taylor et al. 2021)) indicates a larger number of innexin genes are expressed in these cells. This should, at the very least, be acknowledged in the text. The authors also present no evidence that the RNAi actually knocks down the target genes or that it is expressed in the desired cells. Again, in the absence of controls, this should at least be acknowledged.

Response: We thank the reviewer for this important and constructive comment. We agree that more recent studies, including Bhattacharya et al. (2019) and the CeNGEN project (Taylor et al., 2021), indicate that ventral nerve cord motor neurons

express a broader set of innexin genes than originally reported by Altun et al. (2009). We now explicitly acknowledge this in the revised Results.

We also acknowledge that RNAi knockdown efficiency and cell-type specificity were not directly assessed for the innexin RNAi experiments, and this limitation is now explicitly stated in the Results. Nevertheless, disruption of multiple gap junction components did not affect acid-evoked responses, VA/VB responses persisted in neurotransmission-defective mutants, and DA/DB responses were abolished when synaptic transmission was impaired. Collectively, these observations argue against a major contribution of gap-junctional coupling. We now explicitly note this in the revised Results as follows: “Although additional innexins may be expressed in these neurons, and RNAi efficiency and cell-type specificity were not directly assessed, the data do not support a major contribution of gap junctions.”

Line 145: “hours” does not seem to fit the sentence.

Response: Deleted, thanks.

Figure 2e: gtl-1 is incorrectly written as glt-1.

Response: Corrected, thanks.

Figure 2d, e: n values are given for some genotypes but not for all.

Response: Corrected, thanks.

Figure 4: misspelling of rescue.

Response: Corrected, thanks.

Line 349: “pumping rates” requires more explanation for a non-specialist reader.

Likewise, “increased colony-forming units” (or simply “CFUs” in the figure legend) – a better explanation of the experiment would be helpful.

Response: We thank the reviewer for this helpful suggestion. We have revised the legend of Supplementary Fig. 9 to briefly explain both measurements. In the revised Methods, we now describe pharyngeal pumping as the rhythmic contraction and relaxation of the pharynx that drives bacterial ingestion, with pumping rate serving as a quantitative measure of feeding activity. We also clarify that colony-forming units (CFUs) represent the number of viable bacteria recovered from the intestine following worm lysis, serial dilution, and plating on selective media. Detailed experimental procedures are provided in the Methods.